# Effects of *SPI1*-mediated transcriptome remodeling on Alzheimer's disease-related phenotypes in mouse models of Aβ amyloidosis

Byungwook Kim [1,2], Luke Child Dabin [1,2,9], Mason Douglas Tate[2,3,9], Hande Karahan [1,2,9], Ahmad Daniel Sharify [1,2], Dominic J. Acri [1,2,3], Md Mamun Al-Amin [1,2], Stéphanie Philtjens [1,2], Daniel Curtis Smith [1,2,3], H. R. Sagara Wijeratne [1,2,4], Jung Hyun Park[1,2,5,6], Mathias Jucker [7,8] & Jungsu Kim [1,2,3] ✉

*SPI1* was recently reported as a genetic risk factor for Alzheimer's disease (AD) in large-scale genome-wide association studies. However, it is unknown whether *SPI1* should be downregulated or increased to have therapeutic benefits. To investigate the effect of modulating *SPI1* levels on AD pathogenesis, we performed extensive biochemical, histological, and transcriptomic analyses using both *Spi1*-knockdown and *Spi1*-overexpression mouse models. Here, we show that the knockdown of *Spi1* expression significantly exacerbates insoluble amyloid-β (Aβ) levels, amyloid plaque deposition, and gliosis. Conversely, overexpression of *Spi1* significantly ameliorates these phenotypes and dystrophic neurites. Further mechanistic studies using targeted and single-cell transcriptomics approaches demonstrate that altered *Spi1* expression modulates several pathways, such as immune response pathways and complement system. Our data suggest that transcriptional reprogramming by targeting transcription factors, like *Spi1*, might hold promise as a therapeutic strategy. This approach could potentially expand the current landscape of druggable targets for AD.

Human genetic studies recently identified *SPI1* (encoding PU.1) as a risk factor for Alzheimer's disease (AD)[1–3]. PU.1 is a transcription factor that is mainly expressed by myeloid lineage cells, including microglia[4,5]. In vitro cell culture studies have demonstrated that *Spi1* regulates the expression of AD-associated genes involved in the phagocytic activity and immune response of microglia[1,6]. Because these in vitro systems cannot capture the inherent complexity of the brain and the crosstalk between different cell types, it is unknown how modulation of *Spi1* levels may affect AD pathology in vivo. In addition, two somewhat opposing hypotheses have been proposed based on cell culture

[1]Department of Medical & Molecular Genetics, Indiana University School of Medicine, Indianapolis, IN 46202, USA. [2]Stark Neurosciences Research Institute, Indiana University School of Medicine, Indianapolis, IN 46202, USA. [3]Medical Neuroscience Graduate Program, Indiana University School of Medicine, Indianapolis, IN 46202, USA. [4]Department of Biochemistry and Molecular Biology, Indiana University School of Medicine, Indianapolis, IN 46202, USA. [5]Department of Psychological and Brain Sciences, Indiana University, Bloomington, IN 47405, USA. [6]Program in Neuroscience, Indiana University, Bloomington, IN 47405, USA. [7]German Center for Neurodegenerative Diseases (DZNE), Tübingen, Germany. [8]Department of Cellular Neurology, Hertie Institute for Clinical Brain Research, University of Tübingen, Tübingen, Germany. [9]These authors contributed equally: Luke Child Dabin, Mason Douglas Tate, Hande Karahan. ✉e-mail: jk123@iu.edu

studies. Earlier studies demonstrated that downregulation of *Spi1* in the BV2 microglial cell line decreased the expression of proinflammatory genes while increasing the transcription of genes essential in lipid metabolism[1]. Because proinflammatory cytokines and dysregulation of lipid metabolism are linked to the pathogenesis of AD, their findings suggest that *Spi1*-knockdown may have beneficial effects on certain AD pathologies[1]. On the contrary, *Spi1*-knockdown also decreases microglial phagocytic activity in vitro, which may lead to an increase in the levels of toxic protein aggregates and dead cell debris[1,7]. These opposing hypotheses on the roles of *Spi1* in microglial activity represent its complex role in AD pathogenesis and the intrinsic difficulty in translating cell culture work into in vivo models. Therefore, it is important to determine the effect of modulating *Spi1* levels on AD-related phenotypes under in vivo conditions.

In this work, we addressed this knowledge gap by utilizing both *Spi1*-knockdown and *Spi1*-overexpression mouse models. We crossbred each mouse model with amyloid-β (Aβ) amyloidosis mouse models to investigate the effects of loss- or gain-of-function of *Spi1* on major AD phenotypes in vivo. Our data demonstrate that *Spi1* knockdown exacerbates multiple pathological hallmarks, including Aβ aggregation, amyloid plaque accumulation, and gliosis. Conversely, *Spi1* overexpression offers significant protection against these phenotypes. To further elucidate the underlying mechanism by which *Spi1* regulates AD-related pathology, we utilized targeted and single-cell transcriptomics approaches and found that several immune pathways relevant to AD pathogenesis are regulated by *Spi1*. Collectively, our findings suggest that activating SPI1 or its downstream pathways may have the potential to protect against Aβ and other pathology.

## Results

### Knockdown of *Spi1* increases Aβ aggregation and amyloid deposition

To investigate the role of *SPI1* in the pathological features of AD, we bred *Spi1*-knockdown mice with APP/PS1 mice[8]. Using an SD of 80, an effect size of 40% for insoluble Aβ$_{42}$ levels, a power of 0.8, and $P < 0.05$, we aimed to generate at least 11 mice per genotype. The conventional *Spi1*-knockout mice have early lethality phenotype[9]. Therefore, we decided to utilize a *Spi1*-knockdown mouse model in which the −14 kb upstream regulatory element of the *Spi1* was deleted[10]. Because homozygous *Spi1*-knockdown mice also become moribund from T-cell lymphoma and acute myeloid leukemia starting at 3 months of age[10], we utilized only *Spi1* heterozygous knockdown (*Spi1*$^{+/-}$;APP/PS1) mice and wild-type for *Spi1* (*Spi1*$^{+/+}$;APP/PS1) mice. Before crossing with APP/PS1 mice, we validated the *Spi1*-knockdown mouse model by measuring *Spi1* expression in the brain. The levels of *Spi1* mRNA and protein were decreased by 35.1% and 40.7%, respectively, in the cortices of *Spi1*-knockdown mice compared to littermate wild-type mice (Supplementary Fig. 1a, b).

First, we measured the levels of insoluble Aβ$_{40}$ and Aβ$_{42}$ in the brains of 4-month-old *Spi1*$^{+/+}$;APP/PS1 and *Spi1*$^{+/-}$;APP/PS1 mice using the Meso Scale Discovery (MSD) Aβ electrochemiluminescence assay (Fig. 1a–d). We selected 4 months of age because we observed that this particular APP(Swe)/PS1(L166P) mouse model starts having a considerable amount of Aβ aggregation in the hippocampus after 3–4 months of age[11]. *Spi1*-knockdown significantly increased the levels of insoluble Aβ$_{40}$ and Aβ$_{42}$ in both cortex and hippocampus. In the cortex, insoluble Aβ$_{40}$ and Aβ$_{42}$ levels were increased by 1.71- and 1.48-fold in *Spi1*$^{+/-}$;APP/PS1 mice compared with *Spi1*$^{+/+}$;APP/PS1 mice, respectively (Fig. 1a, b). Likewise, the levels of insoluble Aβ$_{40}$ and Aβ$_{42}$ were increased by 1.89- and 1.93-fold in the hippocampus, respectively (Fig. 1c, d). A previous report demonstrated no sex-dependent difference in the phenotypes of APP(Swe)/PS1(L166P) mice[8]. Just in case there is any unexpected sex difference in the setting of *Spi1*-knockdown, we also analyzed our data from male and female mice separately. The knockdown of *Spi1* had no differential sex effect on Aβ

accumulation (Supplementary Fig. 2a–d). Moreover, there was a significant correlation between Aβ$_{40}$ and Aβ$_{42}$ levels in both cortex and hippocampus of *Spi1*$^{+/-}$;APP/PS1 and *Spi1*$^{+/+}$;APP/PS1 mice (Supplementary Fig. 2e, f).

Next, we assessed amyloid plaque deposition by immunostaining the brain sections with the Aβ-specific antibody 82E1, which detects both diffuse and fibrillar Aβ (Fig. 1e). Consistent with the increased levels of Aβ peptides in *Spi1*$^{+/-}$;APP/PS1 mice, we detected a 1.76- and 2.59-fold increase in amyloid plaque burden in the cortex and hippocampus, respectively, in *Spi1*$^{+/-}$;APP/PS1 mice compared to *Spi1*$^{+/+}$;APP/PS1 mice (Fig. 1f, g). Similarly, the number of Aβ plaques were significantly increased in the cortex and hippocampus of *Spi1*-knockdown mice (Fig. 1h, i). In addition, the overall size of Aβ plaques was enlarged (Supplementary Fig. 2g). To further characterize the nature of the deposited plaques, we stained the brain sections with X-34 dye[12], which detects a fibrillar form of amyloid deposits with β-sheet structures (Fig. 1j). Fibrillar plaque deposition was increased by 1.56- and 1.59-fold in the cortical and hippocampal areas of *Spi1*$^{+/-}$;APP/PS1 mice, respectively (Fig. 1k, l). Consistent with these results, the number of fibrillar plaques was also significantly increased in the cortex and hippocampus of *Spi1*$^{+/-}$;APP/PS1 mice (Fig. 1m, n). To determine whether *Spi1* knockdown affects only certain form of amyloid, such as fibrillar versus diffuse plaques, we analyzed the ratio of fibrillar plaque to total plaque load. No difference in that ratio was observed (Supplementary Fig. 2h, i). This finding suggests that *Spi1* knockdown did not have any preferential effect on fibrillar or diffuse plaques, but rather increased the overall amyloid plaque road. Taken together, these data indicate that *Spi1*-knockdown increases Aβ accumulation and plaque deposition in the brain.

To understand a possible mechanism by which *Spi1*-knockdown increases Aβ accumulation, we assessed the levels of proteins involved in Aβ generation. We measured the levels of APP, its cleaving enzyme β-secretase 1 (BACE1)[13], and the C-terminal fragment of APP (β-CTF)[14] by Western blotting to determine whether *Spi1*-knockdown affects APP processing. We observed no difference in APP, BACE1, and β-CTF protein levels in *Spi1*$^{+/-}$;APP/PS1 mice compared to *Spi1*$^{+/+}$;APP/PS1 mice (Supplementary Fig. 3a–d), suggesting that *Spi1*-knockdown does not affect APP processing.

### Exacerbation of microgliosis and astrogliosis by *Spi1* knockdown

The reactive gliosis, encompassing microgliosis and astrogliosis, are prevalent neuropathological hallmarks observed in the brains of both human AD and mouse models of Aβ amyloidosis[8,15,16]. Therefore, we further evaluated whether *Spi1*-knockdown affects both microgliosis and astrogliosis in our Aβ amyloidosis mouse model. We stained brain sections with antibodies against ionized calcium-binding adapter molecule 1 (IBA1) and glial fibrillary acidic protein (GFAP) to assess microglial activation (Fig. 2a) and astrogliosis (Fig. 2c), respectively, in the *Spi1*$^{+/-}$;APP/PS1 mice and *Spi1*$^{+/+}$;APP/PS1 mice. In *Spi1*$^{+/-}$;APP/PS1 mice, the % of cortical area covered by IBA1- and GFAP-positive cells was increased by 1.69- and 1.79-fold, respectively, compared to *Spi1*$^{+/+}$;APP/PS1 mice (Fig. 2b, d). Collectively, these data demonstrate that reduction in *Spi1* level exacerbates microgliosis and astrogliosis in a mouse model of Aβ amyloidosis.

### Overexpression of *Spi1* decreases Aβ accumulation and plaque burden

Given that *Spi1*-knockdown exacerbates amyloid pathology, we wanted to determine whether *Spi1*-overexpression can reduce Aβ accumulation and plaque deposition. To investigate the effects of *Spi1*-overexpression in the mouse brain, we bred *Spi1*-overexpressing transgenic mice (*Spi1*$^{Tg/0}$) with the 5XFAD, Aβ amyloidosis mouse model[16]. Using an SD of 120, an effect size of 40% for insoluble Aβ$_{42}$ levels, a power of 0.8, and $P < 0.05$, we aimed to generate at least 9

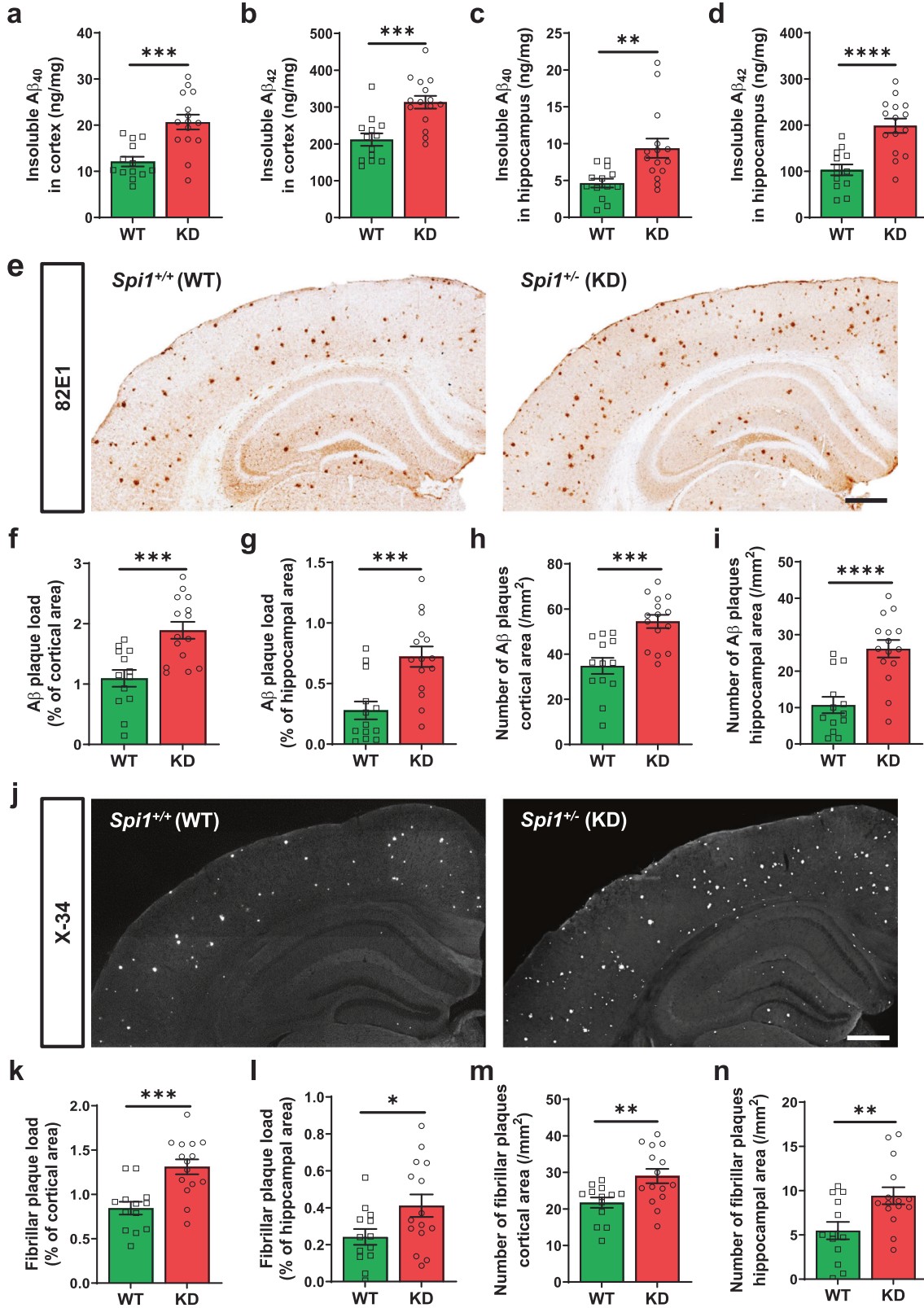

mice per sex for each genotype. Before crossing with 5XFAD mice, we assessed *Spi1* mRNA levels in the brains of *Spi1^Tg/0* mice. The levels of *Spi1* mRNA and protein were significantly increased by 1.41- and 1.54-fold, respectively, in *Spi1^Tg/0* mice relative to littermate controls (Supplementary Fig. 1c, d).

Next, we measured the extent of Aβ accumulation in cortical and hippocampal tissues of *Spi1*-overexpressing 5XFAD (*Spi1^Tg/0*;5XFAD)

mice (Fig. 3a–d). *Spi1*-overexpression significantly decreased the levels of insoluble Aβ40 and Aβ42 by 71.5% and 52.9%, respectively, in the cortices of female 5XFAD mice (Fig. 3a). We also determined the extent of Aβ aggregation in the hippocampus of these mice. Levels of Aβ40 and Aβ42 were significantly decreased in the hippocampus by 29.5% and 22.2%, respectively, by *Spi1*-overexpression (Fig. 3b). Similar decreases were also observed in male mice (Fig. 3c, d). Consistent with

**Fig. 1 | Reducing *Spi1* expression increases amyloid plaque deposition in a mouse model of Aβ amyloidosis.** The levels of insoluble $A\beta_{40}$ (**a**, **c**) and $A\beta_{42}$ (**b**, **d**) in the guanidine fraction were measured from the cortex (**a**, **b**) and hippocampus (**c**, **d**) of *Spi1*$^{+/+}$;APP/PS1 and *Spi1*$^{+/-}$;APP/PS1 mice using Meso Scale Discovery (MSD) Aβ electrochemiluminescence assay. **e** Representative images of 82E1 (N-terminal Aβ-specific antibody)-positive plaques in brain sections from *Spi1*$^{+/+}$;APP/PS1 and *Spi1*$^{+/-}$;APP/PS1 mice. *Scale bars*, 500 μm. Quantification of Aβ plaque load (**f**, **g**) and total number of Aβ plaques (**h**, **i**) in the cortical area (**f**, **h**) and hippocampal area

(**g**, **i**) of *Spi1*$^{+/+}$;APP/PS1 and *Spi1*$^{+/-}$;APP/PS1 mice. **j** Representative images of X-34-positive fibrillar plaques in brain slices from *Spi1*$^{+/+}$;APP/PS1 and *Spi1*$^{+/-}$;APP/PS1 mice. *Scale bars*, 500 μm. Quantification of fibrillar plaque load (**k**, **l**) and total number of fibrillar plaques (**m**, **n**) in the cortical area (**k**, **m**) and hippocampal area (**i**, **n**) of *Spi1*$^{+/+}$;APP/PS1 and *Spi1*$^{+/-}$;APP/PS1 mice. All values are mean ± SEM. *$p < 0.05$, **$p < 0.01$, ***$p < 0.001$, and ****$p < 0.0001$ (unpaired, two-tailed *t*-test; WT, $n = 13$ ($n = 8$ males, $n = 5$ females) for *Spi1*$^{+/+}$;APP/PS1; KD, $n = 15$ ($n = 9$ males, $n = 6$ females) for *Spi1*$^{+/-}$;APP/PS1. Source data are provided as a Source data file.

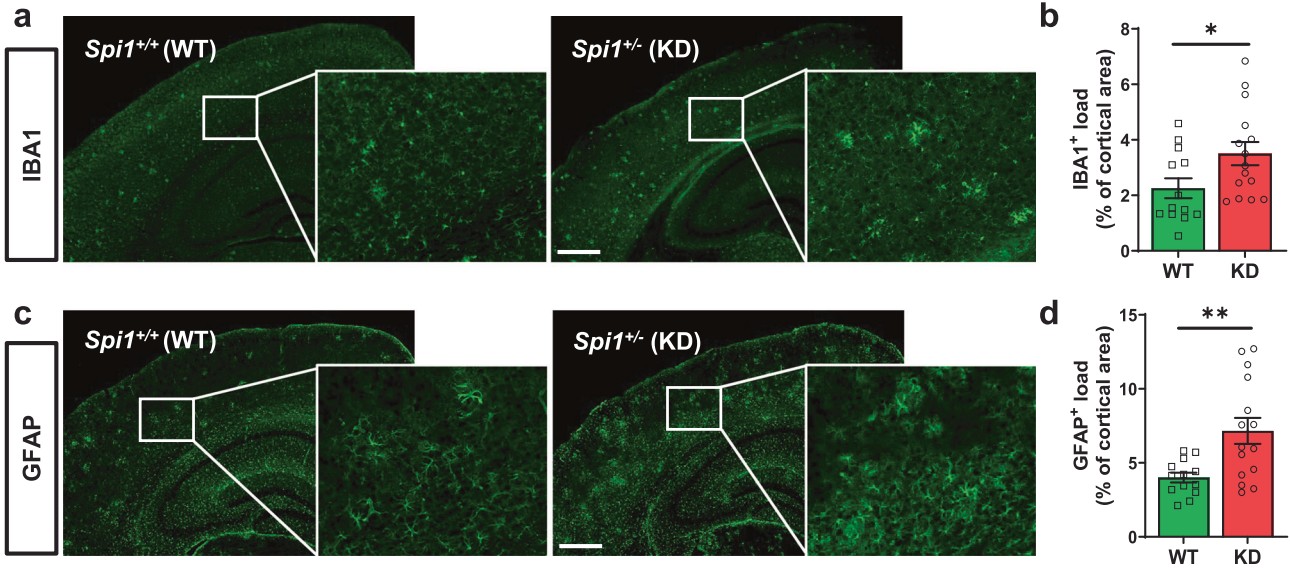

**Fig. 2 | *Spi1*-knockdown increases microgliosis and astrogliosis in mouse model of Aβ amyloidosis. a** Representative images of IBA1-positive cells in brain slices from *Spi1*$^{+/+}$;APP/PS1 and *Spi1*$^{+/-}$;APP/PS1 mice. *Scale bars*, 500 μm. **b** Quantification of the IBA1-positive cell load in the cortical area. **c** Representative images of GFAP-positive cells in brain slices from *Spi1*$^{+/+}$;APP/PS1 and *Spi1*$^{+/-}$;APP/PS1 mice. *Scale*

*bars*, 500 μm. **d** Quantification of the GFAP-positive area in the cortex. All values are mean ± SEM. *$p < 0.05$ and **$p < 0.01$ (unpaired, two-tailed *t*-test; WT, $n = 13$ ($n = 8$ males, $n = 5$ females) for *Spi1*$^{+/+}$;APP/PS1; KD, $n = 15$ ($n = 9$ males, $n = 6$ females) for *Spi1*$^{+/-}$;APP/PS1). Source data are provided as a Source data file.

the reported phenotypes of the 5XFAD model[17], we found that Aβ accumulation in females was almost double that of males. *Spi1*-overexpression significantly decreased Aβ accumulation consistently in both sexes (Fig. 3a–d). The levels of $A\beta_{40}$ strongly correlated with $A\beta_{42}$ levels in each brain region (Supplementary Fig. 4a–d).

To determine the extent of amyloid plaque deposition, we stained the brain sections with Aβ antibody and quantified the area covered by Aβ and the number of plaques per mouse (Fig. 3e–i). *Spi1*-overexpression significantly decreased Aβ plaque load by 45.3% and 46.8% in the cortical and hippocampal areas of female mice, respectively (Fig. 3f, g). Furthermore, the number of Aβ plaques was also significantly decreased in female *Spi1*$^{Tg/0}$;5XFAD mice compared to *Spi1*$^{+/+}$;5XFAD mice (Fig. 3h, i), accompanied by a further reduction in the overall size of Aβ plaques (Supplementary Fig. 4e). To determine whether fibrillar forms of amyloid were also affected by *Spi1*-overexpression, we assessed the extent of fibrillar plaque deposition by X-34 staining (Fig. 3j–n). Fibrillar plaque burden was significantly decreased by 55.6% and 60.9% in the cortices and hippocampi of female *Spi1*$^{Tg/0}$;5XFAD mice compared to *Spi1*$^{+/+}$;5XFAD mice, respectively (Fig. 3k, l). The number of fibrillar plaques was also significantly decreased (Fig. 3m, n). Consistent with the effect of *Spi1*-knockdown on plaques (Supplementary Fig. 2h, i), the ratio of fibrillar plaque to total plaque load remained unchanged upon *Spi1*-overexpression (Supplementary Fig. 4f, g). To increase the rigor of our study design, we also analyzed male mice separately. *Spi1*-overexpression in male mice significantly attenuated amyloid aggregation, as measured by both anti-Aβ antibody and X-34 dye in both cortex and hippocampus (Supplementary Fig. 5a–f).

Similar to the *Spi1*-knockdown study (Supplementary Fig. 3a–d), we evaluated the protein levels of APP, BACE1, and β-CTF to determine whether *Spi1*-overexpression influences APP processing. *Spi1*-overexpression did not affect APP, BACE1, and β-CTF protein levels in the brains of 5XFAD mice (Supplementary Fig. 3e–h), suggesting that *Spi1*-overexpression did not alter APP processing.

**Amelioration of gliosis by *Spi1* overexpression**

Given that glial activation was increased in the *Spi1*-knockdown study (Fig. 2), we next investigated whether *Spi1*-overexpression inhibits glial activation in our Aβ amyloidosis mouse model. We stained brain sections with IBA1 and GFAP antibodies to assess microgliosis (Fig. 4a) and astrogliosis (Fig. 4c), respectively, in the *Spi1*$^{Tg/0}$;5XFAD mice and *Spi1*$^{+/+}$; 5XFAD mice. *Spi1*$^{Tg/0}$;5XFAD mice significantly decreased the % of cortical area covered by IBA1- and GFAP-positive cells by 46.2% and 54.4%, respectively (Fig. 4b, d), compared to *Spi1*$^{+/+}$;5XFAD mice. Collectively, these data demonstrate that *Spi1*-overexpression reduces gliosis in a mouse model of Aβ amyloidosis.

**NanoString analysis suggests that knockdown and over-expression of *Spi1* affect amyloid pathology via regulating microglial function**

To identify potential pathways by which *SPI1* affects AD-associated phenotypes, we next evaluated the gene expression profile in the cortices of *Spi1*$^{+/-}$;APP/PS1 versus *Spi1*$^{+/+}$;APP/PS1 mice using the NanoString mouse AD panel. This panel covers 770 AD-associated genes implicated in multiple pathways (Supplementary Data 1). We

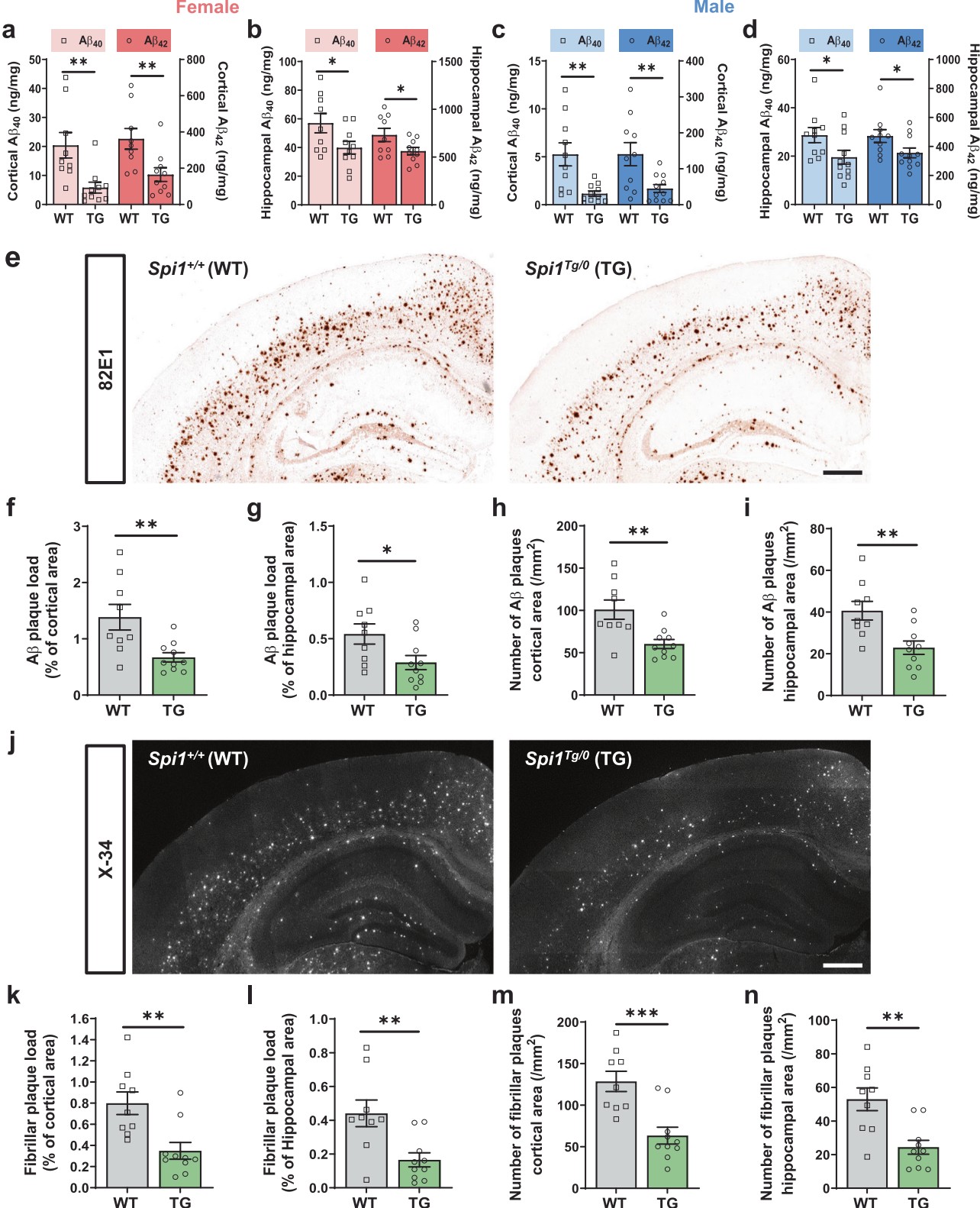

identified 26 differentially expressed genes (DEGs) ($p < 0.05$, Supplementary Data 2). Among these DEGs, we identified *Pard3* and *Hspa1b* to be downregulated and *Tcirg1, Tyrobp,* and *Cd74* to be upregulated in *Spi1+/−*;APP/PS1 mice (Fig. 5a). Pathway enrichment analysis of the DEGs using Enrichr[18] identified the "TYROBP Causal Network, WP3625 (*Tyrobp* and *Tcirg1*)" microglial pathway (Fig. 5b and Supplementary Data 3) and MetaCore analysis revealed the "Immune response_Antigen presentation by MHC class I (*Hsp70,*

*Hspa1a, Cd74,* and *Tyrobp*)" (Fig. 5c and Supplementary Data 4) as the most significantly enriched pathway. We validated those DEGs involved in the "TYROBP Causal Network (*Tyrobp* and *Tcirg1*)" and "Immune response_Antigen presentation by MHC class I (*Hspa1a, Cd74,* and *Tyrobp*)", and DEG such as *Pard3* in the cortex of *Spi1+/−*;APP/PS1 mice and *Spi1+/+*;APP/PS1 mice using qPCR analysis. Consistent with the NanoString data (Fig. 5a), *Tyrobp, Tcirg1,* and *Cd74* mRNA levels were significantly upregulated in *Spi1+/−*;APP/PS1

**Fig. 3 | Overexpression of *Spi1* reduces plaque deposition in a mouse model of Aβ amyloidosis.** The levels of insoluble Aβ$_{40}$ and Aβ$_{42}$ were measured in female (**a**, **b**) and male (**c**, **d**) 5XFAD mice using Meso Scale Discovery (MSD) Aβ assay. Aβ$_{40}$ and Aβ$_{42}$ levels in the guanidine fraction from the cortex (**a**, **c**) and hippocampus (**b**, **d**) of *Spi1*$^{+/+}$;5XFAD and *Spi1*$^{Tg/0}$;5XFAD mice. All values are mean ± SEM. *$p < 0.05$ and **$p < 0.01$ (unpaired, two-tailed *t*-test; WT, $n = 9$ for female *Spi1*$^{+/+}$;5XFAD, $n = 10$ for male *Spi1*$^{+/+}$;5XFAD; TG, $n = 10$ for female *Spi1*$^{Tg/0}$;5XFAD, $n = 11$ for male *Spi1*$^{Tg/0}$;5XFAD). **e** Representative images of 82E1-positive Aβ plaques in brain sections from *Spi1*$^{+/+}$;5XFAD and *Spi1*$^{Tg/0}$;5XFAD mice. *Scale bars*, 500 μm. Quantification of Aβ plaque load (**f**, **g**) and total number of Aβ plaques (**h**, **i**) in the cortical (**f**, **h**) and

hippocampal (**g**, **i**) area of female *Spi1*$^{+/+}$;5XFAD and *Spi1*$^{Tg/0}$;5XFAD mice. All values are mean ± SEM. *$p < 0.05$ and **$p < 0.01$ (unpaired, two-tailed *t*-test; WT, $n = 9$ for *Spi1*$^{+/+}$;5XFAD; TG, $n = 10$ for *Spi1*$^{Tg/0}$;5XFAD). **j** Representative images of X-34-positive fibrillar plaques in brain sections from female *Spi1*$^{+/+}$;5XFAD and *Spi1*$^{Tg/0}$;5XFAD mice. *Scale bars*, 500 μm. Quantification of fibrillar plaque load (**k**, **l**) and total number of fibrillar plaques (**m**, **n**) in the cortical (**k**, **m**) and hippocampal (**l**, **n**) of female *Spi1*$^{+/+}$;5XFAD and *Spi1*$^{Tg/0}$;5XFAD mice. All values are mean ± SEM. **$p < 0.01$ and ***$p < 0.001$ (unpaired, two-tailed *t*-test; WT, $n = 9$ for *Spi1*$^{+/+}$;5XFAD; TG, $n = 10$ for *Spi1*$^{Tg/0}$;5XFAD). Source data are provided as a Source data file.

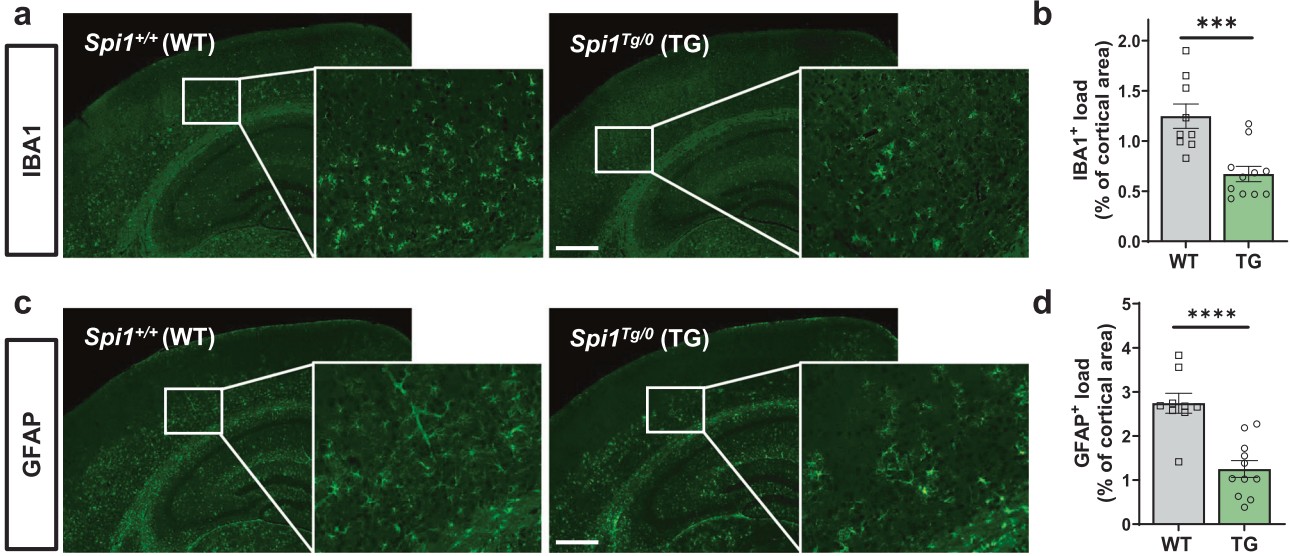

**Fig. 4 | *Spi1*-overexpression decreases microgliosis and astrogliosis in a mouse model of Aβ amyloidosis. a** Representative images of IBA1-positive cells in brain slices from female *Spi1*$^{+/+}$;5XFAD and *Spi1*$^{Tg/0}$;5XFAD mice. *Scale bars*, 500 μm. **b** Quantification of the IBA1-positive cell load in the cortical area. **c** Representative images of GFAP-positive cells in brain slices from female *Spi1*$^{+/+}$;5XFAD and *Spi1*$^{Tg/0}$;

5XFAD mice. *Scale bars*, 500 μm. **d** Quantification of the GFAP-positive cell load in the cortical area. All values are mean ± SEM. ***$p < 0.001$ and ****$p < 0.0001$ (unpaired, two-tailed *t*-test; WT, $n = 9$ for *Spi1*$^{+/+}$;5XFAD; TG, $n = 11$ for *Spi1*$^{Tg/0}$;5XFAD). Source data are provided as a Source data file.

mice compared to *Spi1*$^{+/+}$;APP/PS1 mice, whereas *Hspa1b* and *Pard3* were significantly downregulated (Fig. 5d).

Similar to the *Spi1*-knockdown study, to explore potential pathways that are regulated by *SPI1*-overexpression in the context of AD pathology, we performed transcriptomic profiling in the cortices of *Spi1*$^{Tg/0}$;5XFAD and *Spi1*$^{+/+}$;5XFAD mice using the NanoString mouse AD panel. NanoString analysis revealed significant downregulation of microglial genes, including *Tyrobp, Laptm5, Grn, Ctss, C1qa, Fcer1g,* and *Cyba,* while the *Hspa1b* gene was significantly upregulated in *Spi1*$^{Tg/0}$;5XFAD mice compared to *Spi1*$^{+/+}$;5XFAD mice (Fig. 5e and Supplementary Data 5). To understand the specific biological pathways that are regulated by DEGs in *Spi1*$^{Tg/0}$;5XFAD, we performed Pathway enrichment analysis of DEGs using Enrichr and MetaCore™ bioinformatics approaches and identified the "Microglia pathogen phagocytosis pathway (*C1qa, Fcerg1, Tyrobp, Trem2, Cyba, C1qc*)" with Enrichr (Fig. 5f and Supplementary Data 6) and "TGF-beta signaling (*Itgb5, Tgfbr2, Tgfbr1*)" with MetaCore™ (Fig. 5g and Supplementary Data 7).

Next, we also validated several DEGs involved in the "Microglia pathogen phagocytosis pathway (*C1qa, Fcerg1, Tyrobp, Trem2,* and *Cyba*)", and DEGs such as *Ctss, Laptm5,* and *Hspa1a* in the cortex of *Spi1*$^{Tg/0}$;5XFAD mice and *Spi1*$^{+/+}$;5XFAD mice using qPCR analysis. Consistent with the NanoString data (Fig. 5e), *C1qa, Fcer1g, Tyrobp, Trem2, Cyba, Ctss,* and *Laptm5* mRNA levels were significantly downregulated in *Spi1*$^{Tg/0}$;5XFAD mice compared to *Spi1*$^{+/+}$;5XFAD mice, whereas *Hspa1b* was significantly upregulated (Fig. 5h).

## Integrative analysis of transcriptome reveals that *Spi1* regulates key microglial genes

Next, we performed integrative analyses of DEGs using NanoString data from both *Spi1*-knockdown and *Spi1*-overexpression studies to better understand the molecular pathways regulated by *SPI1*. We utilized a linear model[19] of gene expression against genotype to identify DEGs and then performed analyses of Transcription factor enrichment, Biological Pathways, GO processes, and Process Networks using the combined DEG list (Fig. 6a). Unbiased transcription enrichment analysis of lists of DEGs from both *Spi1*-knockdown and *Spi1*-overexpression studies identified PU.1 as a major transcription factor, validating our linear modeling approach and confirming *Spi1*'s regulatory role in our DEG dataset (Fig. 6b and Supplementary Data 8). Furthermore, Pathway analysis identified "Putative pathways of activation of classical complement system and immune response" (Fig. 6c and Supplementary Data 9) as one of the most significantly enriched pathways. Additionally, GO processes and Network analyses identified enrichment of the "Immune system process and immune response" (Supplementary Fig. 6a and Supplementary Data 10) and "Protein folding, cytoskeleton, reproduction, immune response, phagocytosis" (Supplementary Fig. 6b and Supplementary Data 11), respectively.

A total of 63 DEGs were identified in both datasets (Fig. 6d) and, notably, 6 of these were differentially expressed in opposite directions (Fig. 6e). To determine the potential protein-protein interactions

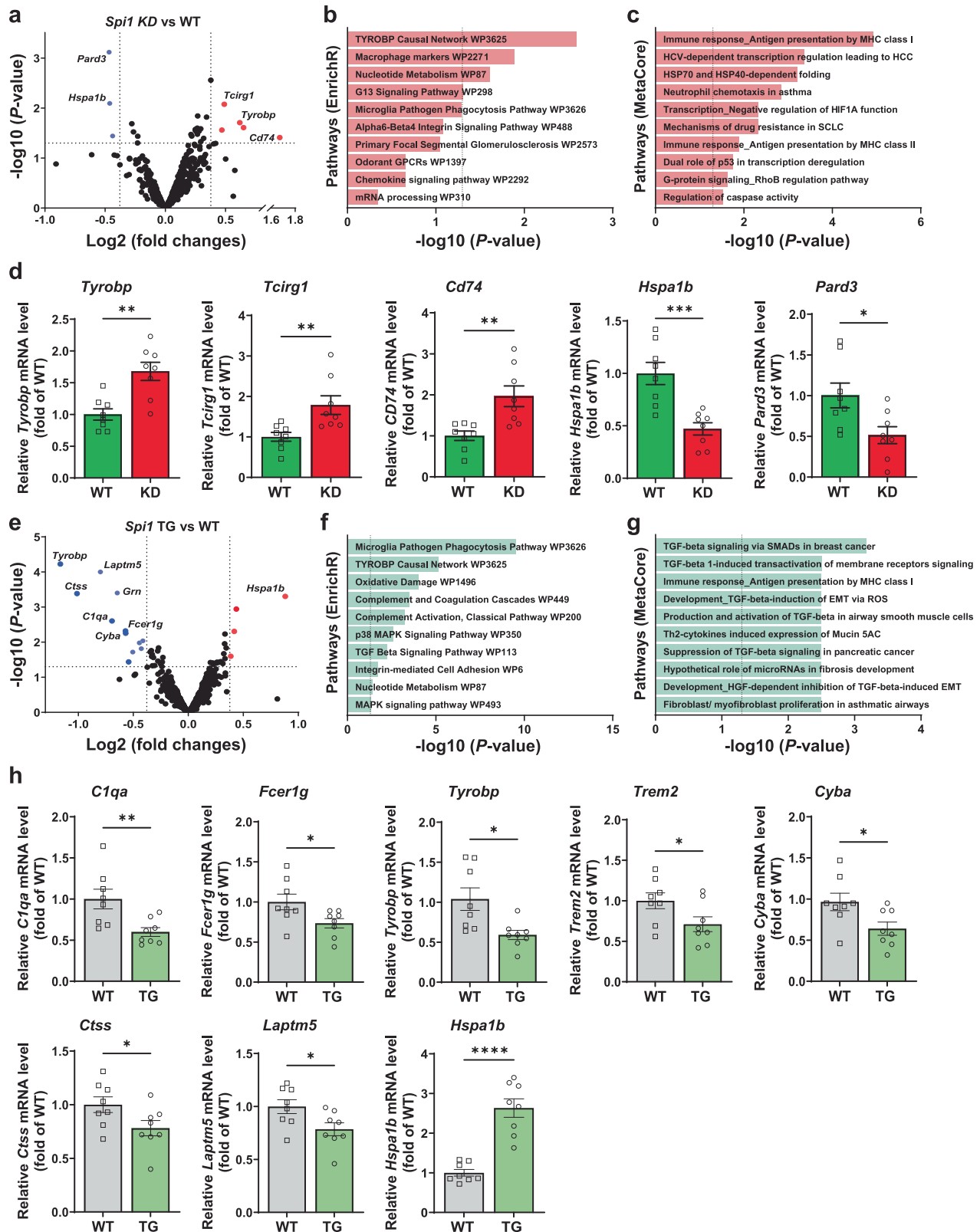

between 63 DEGs and *Spi1*, we performed an interaction analysis using the STRING database (Fig. 6f). This analysis further suggested the involvement of *SPI1* in microglial phagocytosis and the inflammatory response by revealing the interactions between *Spi1* and microglial phagocytosis genes (*Tyrobp, Laptm5, Fcer1g* and *Trem2*), complement genes (*C1qa* and *C1qc*), and inflammatory response genes (*CD74, C1qa, C1qc, Ctss,* and *Trem2*). Integrative pathmap analysis using both lists of

DEGs further demonstrated the interaction between *Spi1* and the immune response and complement system (Fig. 6g).

## *Spi1* modulates microglial Aβ clearance function

Our integrative analysis revealed that *SPI1* may affect amyloid pathology through its regulation of key microglial functions (Fig. 6). Interestingly, our dataset revealed a close physical or functional association

**Fig. 5 | Analyses of transcriptomic changes induced by *Spi1*-knockdown and *Spi1*-overexpression in Aβ amyloidosis mouse models. a–c** Differentially expressed genes (DEGs) in cortex of *Spi1*[+/−];APP/PS1 versus *Spi1*[+/+];APP/PS1 mice identified by NanoString analysis and their pathway enrichment analyses. **a** Volcano plot showing DEGs in *Spi1*[+/−];APP/PS1 mice compared to *Spi1*[+/+];APP/PS1 mice (*n* = 3 males per genotype). The red dots represent significantly upregulated genes, and blue dots represent significantly downregulated genes in *Spi1*[+/−];APP/PS1 versus *Spi1*[+/+];APP/PS1 mice (*p* < 0.05, indicated as a horizontal dash line; fold change > ±1.3, indicated as a vertical dash line). The volcano plot shows statistical significance as the -log$_{10}$ *P*-value (y-axis) and the log$_2$ Fold Change (log$_2$FC, x-axis). Pathway enrichment analysis with the functional annotation of DEGs was performed using EnrichR (**b**) and MetaCore (**c**) software. **d**, qPCR analysis of several DEGs in cortex between *Spi1*[+/−];APP/PS1 mice and *Spi1*[+/+];APP/PS1 mice. All values are mean ± SEM. **p* < 0.05, ***p* < 0.01, and ****p* < 0.001 (unpaired, two-tailed *t*-test; WT, *n* = 8 (*n* = 4 males, *n* = 4 females) for *Spi1*[+/+];APP/PS1; KD, *n* = 8 (*n* = 4 males, *n* = 4 females) for *Spi1*[+/−];APP/PS1). **e–h** DEGs in cortex of *Spi1*[Tg/0];5XFAD versus *Spi1*[+/+]; 5XFAD mice identified by NanoString analysis and pathway enrichment analyses. **e** Volcano plot showing DEGs in *Spi1*[Tg/0];5XFAD versus *Spi1*[+/+];5XFAD mice (*n* = 3 females per genotype). The volcano plot shows statistical significance as the -log$_{10}$ *P*-value (y-axis) and the log$_2$FC (x-axis). The red and blue dots represent significantly upregulated and downregulated genes, respectively, in *Spi1*[Tg/0];5XFAD versus *Spi1*[+/+];5XFAD mice (*p* < 0.05, indicated as a horizontal dash line; fold change > ±1.3, indicated as a vertical dash line). Pathway enrichment analysis of DEGs identified using EnrichR (**f**) and MetaCore (**g**) software. **h** qPCR analysis of several DEGs in cortex between females *Spi1*[Tg/0];5XFAD mice and *Spi1*[+/+];5XFAD mice. All values are mean ± SEM. **p* < 0.05, ***p* < 0.01, and *****p* < 0.0001 (unpaired, two-tailed *t*-test; WT, *n* = 8 for *Spi1*[+/+];5XFAD; TG, *n* = 8 for *Spi1*[Tg/0];5XFAD). *P*-values of **b** and **f** were calculated using two-tailed *Fisher's* exact test, while those for **c** and **g** were determined using Metacore algorithms, both with a threshold of significant enrichment as *p* < 0.05 (shown as a vertical dash line for **b**, **c**, **f**, and **g**). Source data are provided as a Source data file.

between the *C1q*, *Trem2*, *Tyrobp*, and *Syk* genes (Fig. 6f, g). Recent studies have reported the crucial role of these genes in microglia response to Aβ and phagocytosis in AD[20,21]. Therefore, we first evaluated a co-localization between plaques and microglia to investigate whether the *Spi1* level could affect microglia response to Aβ in our Aβ amyloidosis mouse models. We stained the brain sections with IBA1 antibody and X-34 amyloid plaque dye (Fig. 7a, c) and quantified the coverage of IBA1[+] microglial area on amyloid plaques. Knocking down *Spi1* significantly reduced microglial response to plaques compared to their littermate control mice (Fig. 7b), whereas overexpressing *Spi1* did not have a significant effect on the coverage of IBA1[+] microglial area on amyloid plaques (Fig. 7d).

To more directly investigate the role of *Spi1* in microglial response to aggregated Aβ, we performed fibrillar Aβ (fAβ) uptake assays with fAβ labeled with a pHrodo Red probe. BV-2 microglial cells were transfected with a *Spi1* siRNA or *Spi1* plasmid to knockdown or overexpression of *Spi1*, respectively. Twenty-four hours post-transfection, we evaluated a time-dependent Aβ uptake level after treatment with pHrodo-labeled fAβ. *Spi1*-knockdown significantly decreased fAβ uptake (Fig. 7e), while *Spi1*-overexpression increased (Fig. 7f). These findings suggest that *Spi1* levels modulate microglial Aβ clearance function, consequently altering microglial responses around amyloid plaques.

### Decrease in dystrophic neurites by *Spi1* overexpression

Accompanied by gliosis, dystrophic neurites are observed surrounding Aβ plaques in AD[22]. Especially, the extent of dystrophic neurite pathology is often used as an important readout of plaque-associated neuronal toxicity[23]. The lysosome-associated membrane protein 1 (LAMP1) surrounds amyloid plaques in human cases of AD[24] and mouse models of AD[25,26]. The LAMP1-immunoreactivity around amyloid plaques originates from axonal dystrophic neurites[25]. Therefore, we evaluated whether *Spi1*-overexpression can affect axonal dystrophic neurites in our mouse model using a LAMP1 antibody. Brain sections were stained with both anti-LAMP1 and anti-Aβ antibodies (Fig. 8a). The extent of LAMP1 immunoreactivity was significantly decreased by 36.3% and 48.0% in the cortical and hippocampal areas of *Spi1*[Tg/0]; 5XFAD compared to *Spi1*[+/+];5XFAD mice, respectively (Fig. 8b, c). In addition, the number of plaque associated LAMP1 clusters (82E1[+] and LAMP1[+]) was significantly decreased in the cortex and hippocampus area of *Spi1*[Tg/0];5XFAD compared to *Spi1*[+/+];5XFAD mice (Fig. 8d, e, respectively). Taken together, our data demonstrate that overexpression of *Spi1* ameliorates plaque-associated neuronal toxicity.

### Single-cell RNA sequencing data demonstrate that over-expression of *Spi1* induces AD-associated and cell-type-specific transcriptional changes

Furthermore, we conducted single-cell RNA sequencing (scRNA-seq) to determine whether *Spi1*-overexpression induces cell-type-specific transcriptional changes. To enrich microglial cell types, we used a gentle enzymatic dissociation method that we optimized[27,28]. Since harsh dissociation methods can cause the aberrant activation of cells, especially glial cells, we performed all steps at 4 °C to minimize this effect. Data from a total of 16,456 cells passed quality control and formed 19 clusters (Fig. 9a). We annotated each cluster using the single-cell Mouse Cell Atlas (scMCA) package for single-cell mapping[29] and confirmed the enrichment of microglia (M) in our dataset. We also detected other cell types, astrocytes (A), endothelial cells (En), erythrocytes (Er), macrophages (Ma), neurons (N), oligodendrocytes (O), and T-cells (T). There were no significant differences in cell populations within each cluster between the two genotypes (Supplementary Fig. 7).

To gain more insight into the potential pathways regulated by *Spi1*-overexpression, we identified DEGs between *Spi1*[Tg/0];5XFAD and *Spi1*[+/+];5XFAD mice for each cell-type cluster (Supplementary Data 12 and Supplementary Fig. 8). Astrocyte clusters 1–2, Neurons, and Microglia clusters 1–10 exhibited significant changes in gene expression caused by *Spi1*-overexpression. After applying a 1.5-fold-change threshold, we identified 16 DEGs in astrocytes, 4 DEGs in neurons, and 27 DEGs in microglia (Supplementary Data 12). In microglia, several genes were differentially expressed in the same direction across multiple microglia clusters. For example, *Camk2a, Camk2n1, H2-D1, Olfml3*, and *Qpct* were upregulated in microglial clusters 4 and 10, while *H2-DMa* was downregulated in microglial clusters 1,2,3,4,5,6, and 9 (Supplementary Data 12).

To further understand the broader meaning of differential gene expression in *Spi1*[Tg/0];5XFAD mice, we performed gene ontology (GO) and KEGG pathway analyses using DEGs from each microglial cluster. Interestingly, the most significantly enriched terms were multiple neurodegenerative disorders (prion disease, Parkinson's disease, Huntington's disease, Amyotrophic lateral sclerosis, and AD) as well as metabolic and immunological terms (Fig. 9b). Similarly, GO analysis identified the enrichment of Biological Process terms that are involved in metabolic functions as well as gene expression, as expected given PU.1 (encoded by *Spi1*)'s role as a transcription factor (Supplementary Data 13). Furthermore, we compared gene expression across clusters with the disease-associated microglia (DAM) and homeostatic microglia gene list[30] and found that microglial clusters 4, 8, and 10 expressed higher levels of DAM genes and lower levels of homeostatic genes compared to other clusters (Supplementary Fig. 10a).

To complement the DEG analysis, we performed a single-cell WGCNA analysis (scWGCNA)[31]. This powerful technique curates modules of co-expressed genes which can be specific to certain cell-type clusters and be differentially preserved/enriched between *Spi1* genotypes. Using scWGCNA, we identified 16 co-expressed gene modules (Fig. 9c). By performing an integrative analysis of scWGCNA data and

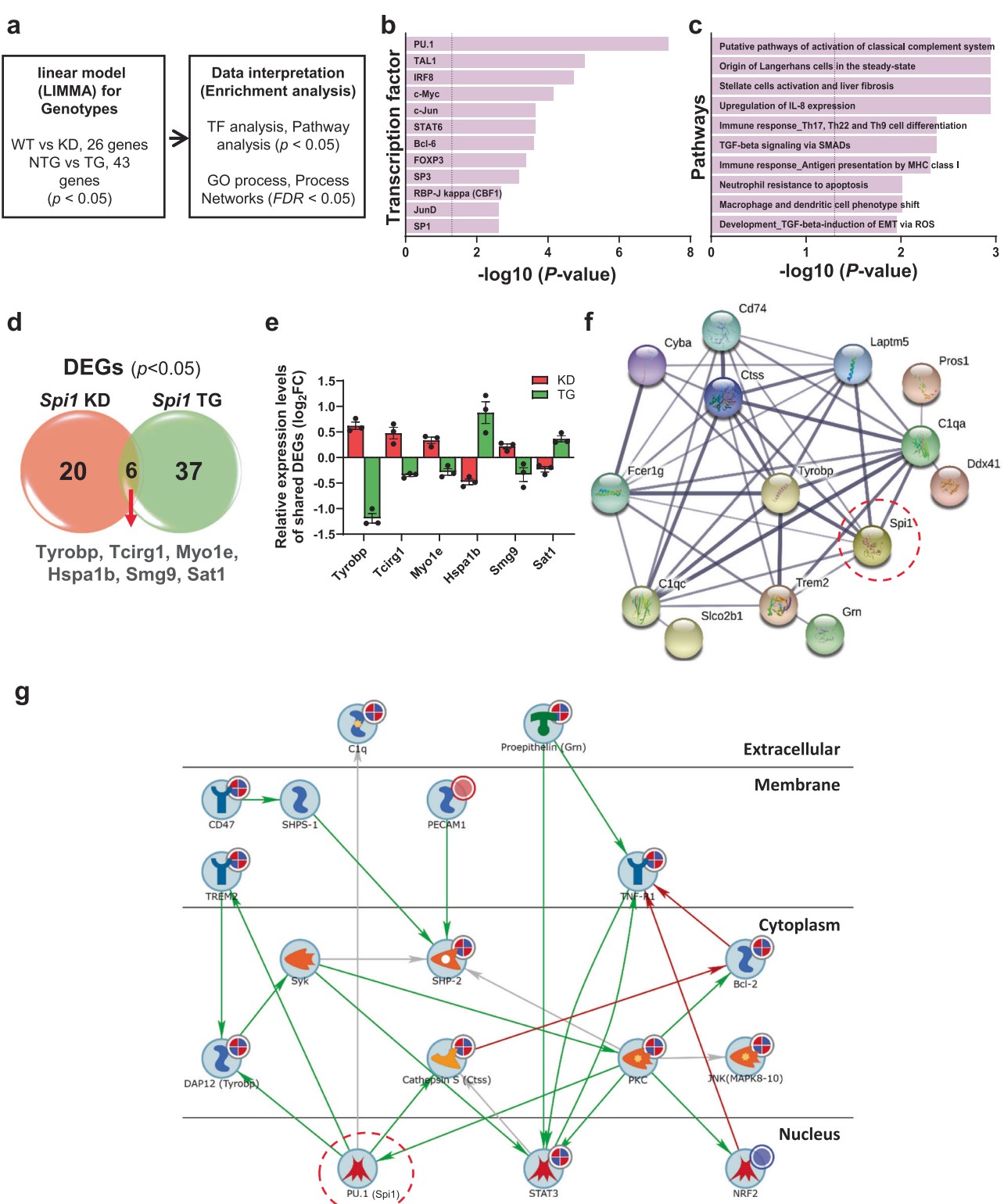

cell-type clustering data, we found that these modules were enriched across different clusters (Fig. 9d and Supplementary Data 14). Strikingly, the light-cyan module was highly enriched across Microglial Cluster 4 and contained hub genes, such as *Lyz2, Cst7, Ccl3*, and *Ccl6*, further reinforcing the DAM-like phenotype of Microglial Cluster 4 (Supplementary Fig. 9a). Comparison of module preservation between *Spi1*[Tg/0];5XFAD and *Spi1*[+/+];5XFAD mice did not reveal any significant difference in density or connectivity for any module (Supplementary Fig. 9b).

Next, we determined whether there were any patterns of communication between the microglial clusters using the CellChat[32] method. In our prior analysis, microglial clusters 4, 8, and 10 expressed higher levels of DAM genes compared to other clusters (Supplementary Fig. 10a). Because microglial cluster 4 had the highest DAM expression, we selected this cluster as our focus for the analysis of microglial cell-cell communication. Furthermore, since this cluster 4 also contained the greatest number of cells compared to the microglial clusters 8 and 10, it also increased the

**Fig. 6 | Integrative analysis of *Spi1*-mediated transcriptomic alterations.** DEGs in *Spi1*$^{+/-}$;APP/PS1 (KD) versus *Spi1*$^{+/+}$;APP/PS1 (WT) and *Spi1*$^{Tg/0}$;5XFAD (TG) versus *Spi1*$^{+/+}$;5XFAD (WT) mice were analyzed with a linear regression model for genotypes. **a** Workflow for LIMMA analysis. Transcription factor enrichment analysis (**b**) and Pathway enrichment analysis (**c**) with combined DEGs were performed using the MetaCore software. *P*-values of **b** and **c** were calculated using Metacore algorithms with a threshold of significant enrichment as $p < 0.05$ (shown as a vertical dash line for **b** and **c**). **d** All DEGs were summarized in a Venn diagram, which identified six shared DEGs across *Spi1*$^{+/-}$;APP/PS1 versus *Spi1*$^{+/+}$;APP/PS1 mice and *Spi1*$^{Tg/0}$;5XFAD versus *Spi1*$^{+/+}$;5XFAD mice. **e** Relative expression levels of six shared DEGs. Quantification data were expressed as log$_2$FC of expression relative to each

control group. All values are mean ± SEM ($n = 3$ per group). **f** A protein-protein interaction network was generated using the STRING database for the combined DEGs (63 genes) and *Spi1*. The data showed interactions with *Spi1* and among all DEGs. Wider lines represent stronger evidence of interactions. **g** Integrative path-map analysis was performed using both sets of DEGs using MetaCore. Up-regulated genes in our dataset are shown with red circles, down-regulated genes are shown with blue circles, and the mixed-signal gene is shown with a red-blue mixed circle in the pathmap. Green arrows between nodes represent activation, while gray arrows represent interaction with no specific direction of effect. DEGs shown are involved in the immune response and complement system. Source data are provided as a Source data file.

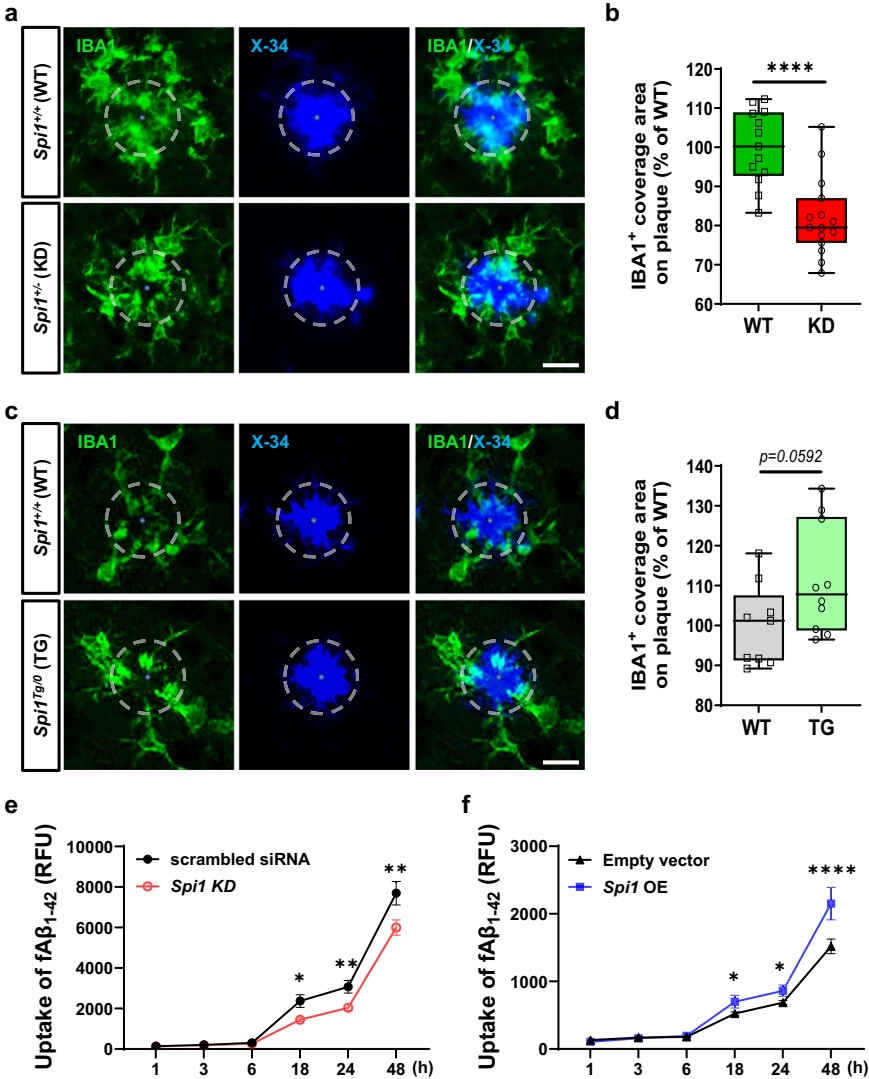

**Fig. 7 | *Spi1*-knockdown reduces microglial response to plaques and Aβ uptake, whereas *Spi1*-overexpression increases them.** Representative images of colocalization between IBA1-positive microglia and X-34-positive fibrillar plaque in brain slices from *Spi1*$^{+/+}$;APP/PS1 and *Spi1*$^{+/-}$;APP/PS1 mice (**a**) or from *Spi1*$^{+/+}$;5XFAD and *Spi1*$^{Tg/0}$;5XFAD mice (**c**). *Scale bars*, 20 μm. **b, d** Quantification of the IBA1-positive cell coverage area on the fibrillar plaque. Within each box, horizontal lines denote median values; boxes extend from the 25th to the 75th percentile of each group's distribution of values. The whiskers are shown from min to max, showing all points. ****$p < 0.0001$ (unpaired, two-tailed *t*-test). For **a** and **b**, WT, $n = 13$ ($n = 8$ males, $n = 5$ females) for *Spi1*$^{+/+}$;APP/PS1; KD, $n = 15$ ($n = 9$ males, $n = 6$ females) for *Spi1*$^{+/-}$;APP/PS1. For **c-d**, WT, $n = 9$ females for *Spi1*$^{+/+}$;5XFAD; TG, $n = 10$ females for *Spi1*$^{Tg/0}$;

5XFAD). **e, f** BV-2 microglial cells were transfected with a *Spi1* siRNA or *Spi1* plasmid to knockdown or overexpression of *Spi1*, respectively. Twenty-four hours post-transfection, cells were incubated with 100 nM of Aβ$_{1-42}$ conjugated with pHrodo Red dye for 48 h. Time-lapse of Aβ uptake level of *Spi1*-knockdown (**e**) or -overexpression (**f**) was compared to each negative control (scrambled siRNA or Empty vector, respectively) under Aβ treatment. The Aβ uptake level in the cells was evaluated using the relative fluorescence unit (RFU) of pHrodo Red emission. All values are mean ± SD. *$p < 0.05$, **$p < 0.01$, and ****$p < 0.0001$ (two-way ANOVA, *Sidak's* multiple comparisons test; $n = 5$ per group). Source data are provided as a Source data file.

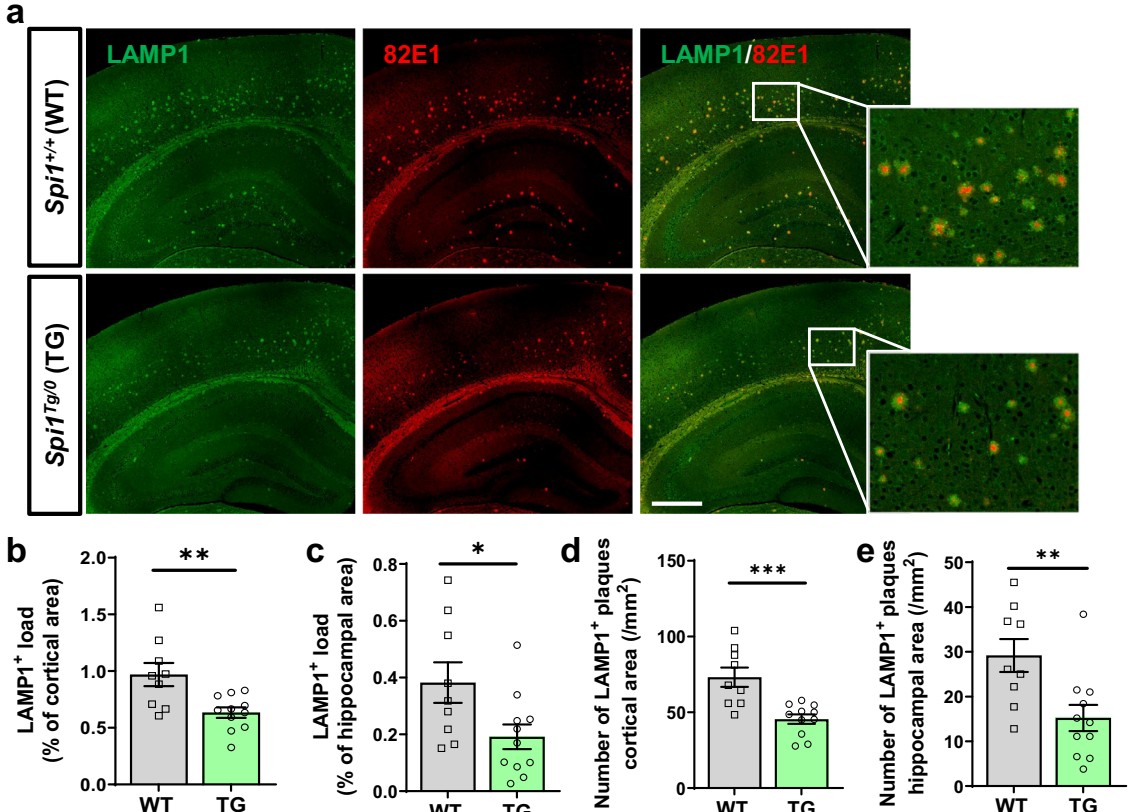

**Fig. 8 | *Spi1*-overexpression decreases axonal dystrophic neurites in a mouse model of Aβ amyloidosis. a** Representative images of brain sections from *Spi1+/+*; 5XFAD and *Spi1Tg/0*;5XFAD mice co-stained with anti-LAMP1 and -Aβ (82E1) antibodies. *Scale bars*, 500 μm. Quantification of the LAMP1-positive load (**b, c**) and number of LAMP1-positive with 82E1-positive plaques (**d, e**) in the cortical (**b, d**) and hippocampal (**c, e**) area of female *Spi1+/+*;5XFAD and *Spi1Tg/0*;5XFAD mice. All values are mean ± SEM. *$p < 0.05$, **$p < 0.01$, and ***$p < 0.001$ (unpaired, two-tailed *t*-test; WT, $n = 9$ for *Spi1+/+*;5XFAD; TG, $n = 11$ for *Spi1Tg/0*;5XFAD). Source data are provided as a Source data file.

statistical power of our analysis. Interestingly, we found that microglial cluster 4 extensively signals to other microglial clusters through a variety of ligands, including *Ccl3* and *Ccl4* (Fig. 9e). As these cytokines are expressed by microglia in response to fibrillar Aβ[33], we decided to further investigate the signaling dynamics of the CCL pathway. Focusing on the CCL pathway ligands and receptors reveals that microglial clusters 4, 8, and 10 signal to all other microglial clusters via this pathway (Fig. 9f). Furthermore, the two principal ligands involved in this signaling are the DAM-associated genes, *Ccl3* and *Ccl4*, while the receptor for these ligands, *Ccr5*, is expressed in all microglial clusters apart from microglia 8 (Supplementary Fig. 10b). To further clarify the direction of communication between ligand and receptor, we focused on the expression of *Ccl3*, *Ccl4*, and *Ccr5*. As expected, *Ccl3* and *Ccl4* are expressed primarily by DAM clusters M4, M8, and M10, while *Ccr5* is expressed in nearly all microglia (Supplementary Fig. 10). Interestingly, *Ccl3* signaling decreases in *Spi1Tg/0*;5XFAD compared to *Spi1+/+*;5XFAD mice, and *Ccr5* is significantly downregulated in microglial clusters M4 and M5 in *Spi1Tg/0*;5XFAD mice (Supplementary Data 15). Therefore, our data suggest that DAM-like microglia use the CCL pathway to communicate with non-DAM microglia in our models.

## Discussion

*SPI1* gene has been identified as a genetic risk factor in recent human AD genetic studies[1–3]. Although its genetic association with AD has been strongly supported, how *SPI1* affects AD pathogenesis remains unclear. Therefore, functional studies using in vivo model systems are warranted to determine the role of *SPI1* in the pathogenesis of AD and future drug discovery efforts.

To address this important knowledge gap, we utilized *Spi1*-knockdown and *Spi1*-overexpression mouse models and crossbred them with Aβ amyloidosis mouse models. We demonstrated that *Spi1*-knockdown significantly increased insoluble Aβ levels and amyloid plaque deposition (Fig. 1). Conversely, *Spi1*-overexpression significantly decreased insoluble Aβ peptides and amyloid plaque deposition (Fig. 3 and Supplementary Fig. 5). These data demonstrate that the expression level of *Spi1* regulates amyloid deposition.

Amyloid-associated gliosis is consistently observed in the brains of AD patients[15] and amyloidosis mouse models[8,16]. Because a previous in vitro study demonstrated that *Spi1*-knockdown decreased, whereas *Spi1*-overexpression increased, the microglial immune response in response to Aβ42 and LPS stimulation[34], we evaluated whether *Spi1* expression affects gliosis in vivo. Contrary to a previous in vitro report, we found that *Spi1*-knockdown increased gliosis (Fig. 2), whereas *Spi1*-overexpression decreased, gliosis (Fig. 4) in amyloidosis mouse models. These apparently conflicting findings between in vitro and in vivo studies could be attributed to the fact that, in addition to regulating immune responses, *Spi1* also has a prominent role in other functions, such as phagocytic activity. Therefore, we further explored whether *Spi1* could affect microglial response to Aβ using our in vivo and in vitro models. We demonstrated that microglial Aβ clearance function altered by *Spi1* level leads to altered plaque-associated microglia (Fig. 7).

Along with gliosis, dystrophic neurites are found surrounding the Aβ plaques. Plaque-associated neuronal toxicity is often assessed by measuring dystrophic neurite pathology[23]. Therefore, we evaluated the effect of *Spi1*-overexpression on the extent of dystrophic neurites by immunostaining with LAMP1 (a marker of dystrophic neurites) and

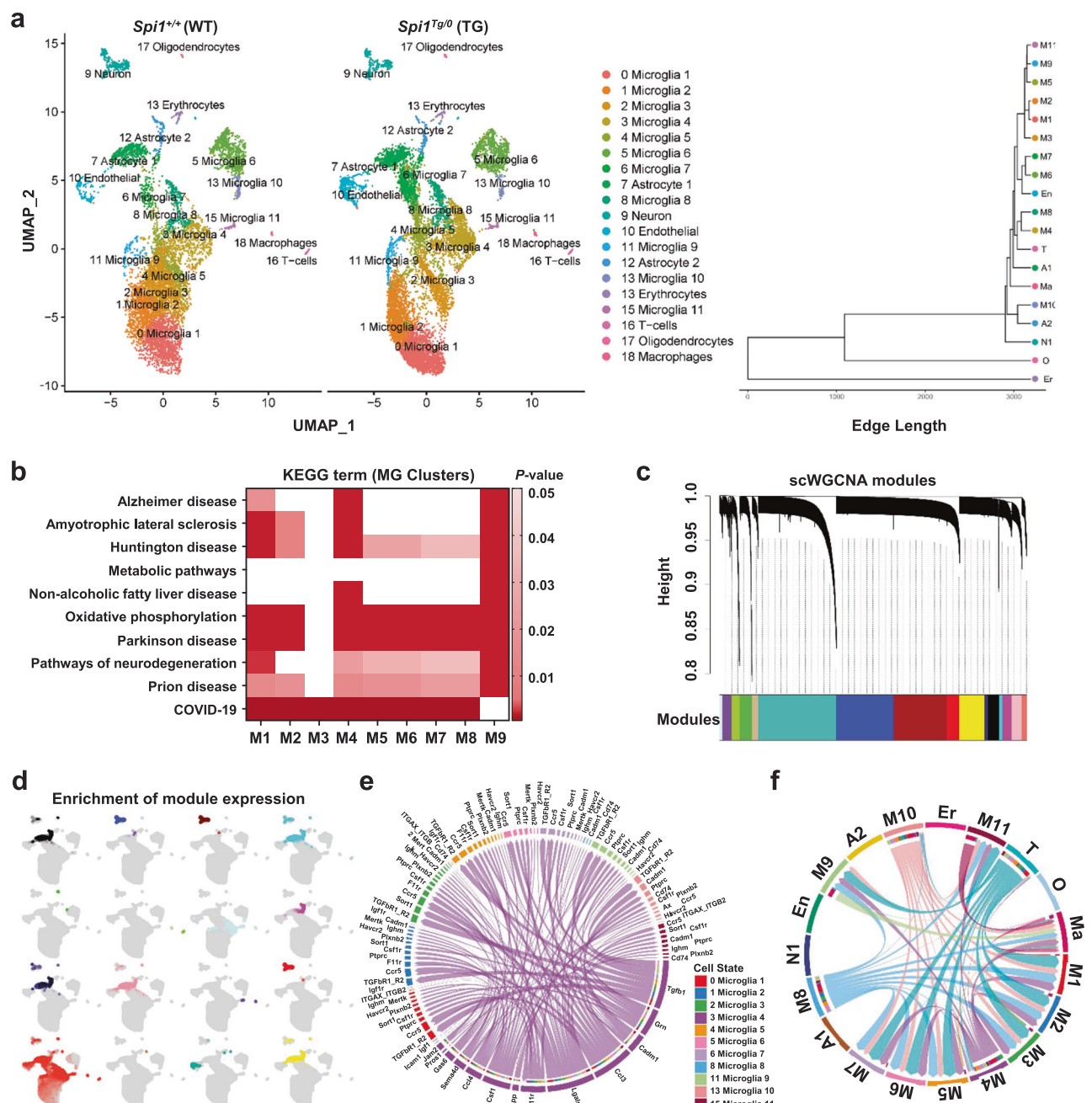

**Fig. 9 | Analyses of transcriptomic changes and cell-type specific gene expression signatures induced by *Spi1*-overexpression in an Aβ amyloidosis mouse model.** a–f scRNA-seq analysis identified 19 clusters of cell types in 4-month-old *Spi1*[+/+];5XFAD and *Spi1*[Tg/0];5XFAD mouse brains. **a** UMAP plots showing the 19 annotated cell type clusters for each genotype. The clustering dendrogram shows cluster relatedness. **b** Heatmap of KEGG terms generated from microglial cluster DEGs. *P*-values of KEGG term enrichment were calculated in Gprofiler2 using a two-tailed *Fisher's* exact test with significant enrichment as *p* < 0.05. Darker colors represent greater term enrichment in each gene list. **c** Sixteen co-expressed gene modules were identified using Single Cell Weighted Gene Network Correlation Analysis (scWGCNA). **d** UMAP plot showing the enrichment of 16 co-expressed gene modules across different clusters. **e** Chord Plot showing that microglial cluster 4

signals to other microglial clusters. The color of each outer band represents the identity of each cell cluster. Inner bands for the cluster 4 microglia represent the identity of each cell cluster that expresses a particular receptor or an interacting protein for the ligand expressed in the cluster 4 microglia. The color of these inner bands is based on the cell cluster targeted by the cluster 4 microglia. **f** CCL Signaling Chord Plot showing CCL pathway signaling between microglial clusters. The color of the outer bands and chords represents the identity of the cell cluster (M1-11 = microglia, A1-2 = astrocytes, En = endothelial cells, Er = erythrocytes, Ma = macrophages, N1 = neurons, O = oligodendrocytes, and T = T-cells). The color of the inner bands is based on the identity of the cell cluster targeted. Source data are provided as a Source data file.

Aβ antibodies. Intriguingly, we observed that *Spi1*-overexpression significantly reduced plaque-associated LAMP1 clusters (82E1[+] and LAMP1[+]) (Fig. 8). Taken together, our data suggest that *Spi1*-over-expression decreases the inflammatory response near amyloid plaques and ameliorates plaque-associated neuronal toxicity.

To gain insight into the potential biological pathways that are regulated by *Spi1*, we performed AD-targeted transcriptomic and single-cell transcriptomic analyses with our mouse models (Figs. 5, 6, and 9). In our targeted transcriptomic analyses, we identified 26 and 43 DEGs in *Spi1*-knockdown and *Spi1*-overexpression studies,

respectively. Among the DEGs from the *Spi1*-knockdown study (Fig. 5a), *Cd74* was significantly upregulated in response to *Spi1*-knockdown. *CD74* was reported to be upregulated in microglia and neurofibrillary tangles in AD patients compared with age-matched controls[35]. In addition, *Pard3* was significantly downregulated by *Spi1*-knockdown. Previous studies demonstrated that the expression of *PARD3* was decreased in AD patients, and knockdown of *Pard3* increased intracellular Aβ accumulation in vitro[36,37]. These findings and their correlation with human AD pathology strengthen the association between genes affected by *Spi1*-knockdown and the increased amyloid pathology in vivo. In the *Spi1*-overexpression study (Fig. 5d), we found the downregulation of multiple microglial genes, such as *Tyrobp*, *Laptm5*, *Grn*, *Ctss*, *C1qa*, *Fcer1g*, and *Cyba*. Notably, *Tyrobp* was upregulated in *Spi1*-knockdown mice (Fig. 5a). Previous studies have shown that a reduction of *Tyrobp* has neuroprotective effects on AD pathology[38,39]. Interestingly, it was reported that PU.1 can bind to the *Tyrobp* gene promoter to activate *Tyrobp* expression[40]. However, in our studies, the expression of *Tyrobp* was not altered in the same direction as reported[40]. It is possible that *Tyrobp* expression might be altered by other proteins that are regulated by *Spi1* in amyloid pathology. Additionally, we found that *Hspa1b* was downregulated by *Spi1*-knockdown (Fig. 5a), whereas it was upregulated by *Spi1*-overexpression (Fig. 5d). It was reported that *Hspa1b* protects against Aβ$_{42}$-induced memory impairments in a *Drosophila* model of AD[41]. Taken together, our results demonstrate that *Spi1* may regulate multiple genes and pathways that are related to AD.

Likely due to PU.1's role as a pioneer transcription factor[42], up- or downregulation of *Spi1* expression in Aβ amyloidosis mouse models altered the expression of multiple genes. It is unlikely that one particular downstream gene is responsible for the phenotypes that we observed. Instead of trying to understand the mechanisms from a reductionist perspective, we think the synergistic effects of multiple downstream genes should be considered. The significant association of these gene lists with metabolism, immune function, and multiple neurodegenerative diseases further establishes the role of *SPI1* in AD-related pathology. Instead of targeting one of these genes, targeting the transcription factor PU.1 could be an effective therapeutic strategy for AD. Interestingly, among different classes of drug target proteins, proteins that directly regulate transcription had one of the highest success rates for drug approval, according to a critical evaluation of all bioactive molecules with drug-like properties in ChEMBL data[43]. One critical concept to keep in mind here is that the targets of transcription factors are not randomly selected. These target genes are almost always within the same or similar pathways that need to be co-regulated in a coordinated manner[43,44]. Therefore, transcription factors could be effective therapeutics for complex diseases because they have a coordinated effect on multiple downstream target genes that need to be co-regulated to restore tissue homeostasis. This is particularly relevant to diseases with complex polygenic risk factors, such as AD. Our transcriptomics data from *Spi1*-knockdown and *Spi1*-overexpression models collectively identified dysregulation of multiple genes related to AD and/or key microglial functions. Importantly, chromatin immunoprecipitation sequencing analysis demonstrated that PU.1 can bind to regulatory elements of multiple AD-associated genes, including *TYROBP*, *TREM2*, *CD33*, *MS4A4A*, and *ABCA7*, and thereby regulate their expression[1]. Therefore, targeting *Spi1* can serve as a combination therapy by restoring the expression of many dysregulated AD-associated genes and their functions. Furthermore, drug discovery studies have suggested that a target with human genetic evidence has a higher translational potential compared to those without genetic evidence[45]. As we mentioned earlier, recent GWAS studies have identified *SPI1* as a genetic risk factor for AD[1–3]. These strong genetic associations further increase the potential of *SPI1* as a therapeutic target in AD. Among the genetic variants linked to AD at the *SPI1* locus, the SNP rs1057233 is known as a protective allele and

was associated with lower *SPI1* expression in myeloid cells[1]. This intriguing inverse correlation between the protective allele and *SPI1* mRNA level suggests that reducing SPI1 expression might potentially offer protection against certain AD phenotypes. On the surface, this hypothesis appears to conflict with our functional data, which showed that amyloid pathology was exacerbated upon knockdown of *Spi1* (Figs. 1 and 2), whereas overexpressing *Spi1* significantly ameliorated these phenotypes (Figs. 3, 4, and 8). This apparent conflict might be explained by the difference in cell types. The rs1057233 SNP is associated with reduced *SPI1* mRNA levels in peripheral immune cells (monocytes and macrophages)[1]. Interestingly, a recent study demonstrated that *SPI1* has a low correlation in gene expression between peripheral monocytes and brain microglia[46]. Therefore, to directly address whether the *SPI1* variant contributes to the disease onset by a loss or gain of function mechanism, it would be necessary to use a mouse model harboring the *SPI1* variant in future studies.

Taken together, our findings demonstrate that *SPI1* plays an important role in amyloid and other pathologies relevant to AD. We identified potential mechanisms for further study that may lead to a more comprehensive understanding of how differential *SPI1* expression may alter crucial microglial functions, including regulation of the immune response, complement system, and phagocytic clearance in AD pathology. Since GWAS-identified variants are identified based on their association with disease risk rather than progression, it is still possible that lower expression of *SPI1* might be protective in later disease stages.

In this study on the functional effect of modulating *SPI1* levels on AD pathology in vivo, we demonstrate that *Spi1* knockdown significantly exacerbates amyloid and other associated pathologies, whereas *Spi1* overexpression ameliorates these phenotypes. This bidirectional effect further strengthens *SPI1*'s potential for therapeutic interventions. It has been demonstrated that drugs targeting genes with bidirectional effects have a higher probability of being approved in the clinic[44]. Collectively, our findings may inform future research into the role of *SPI1* in the pathogenesis of AD, potentially paving the way for further exploration of its therapeutic potential for early-stage AD.

## Methods
### Animals
All mice were maintained under a 12-h light/dark cycle in a temperature-controlled room with free access to food and water. All animal studies were approved and performed in compliance with the guidelines of the Institutional Animal Care and Use Committee of the Indiana University School of Medicine (Protocol ID: 21149).

***Spi1*-knockdown study.** Hemizygous APP/PS1-21 (APP/PS1) transgenic mice on a C57BL6 background containing mutant human APP (Swedish mutations, KM670/671NL) and mutant human PSEN1 (L166P mutation)[8] were crossed with *Spi1*$^{-/-}$ mice on a C57BL6 background (https://www.jax.org/strain/006083; The Jackson Laboratory)[10] to produce *Spi1*$^{+/-}$;APP/PS1 mice. *Spi1*$^{+/-}$;APP/PS1 mice were bred with *Spi1*$^{+/-}$ mice to produce *Spi1*$^{+/+}$;APP/PS1 or *Spi1*$^{+/-}$;APP/PS1 mice. Tissue from four-month-old *Spi1*$^{+/+}$;APP/PS1 (*Spi1* wild-type, WT) and *Spi1*$^{+/-}$;APP/PS1 (*Spi1* knockdown, KD) mice, littermate controls, and both sexes were used. Because it has been known that there is no sex effect in this experimental group[8], male and female mice were combined for statistical analyses.

***Spi1*-overexpression study.** Hemizygous 5XFAD transgenic mice on a C57BL/6SJL background containing mutant human APP (Swedish mutation, KM670/671NL; Florida mutation, I716V; and the London mutation, V717I; and mutant human PS1 (M146L, L286V mutations) (https://www.jax.org/strain/006554)[16] were crossed with *Spi1* transgenic mice on a B6;FVB background (https://www.jax.org/strain/

006147)[47] to generate *Spi1*[+/+];5XFAD (*Spi1* wild-type, WT) and *Spi1*[Tg/0];5XFAD (*Spi1* transgenic, TG) mice. Littermate controls and both sexes were used. Because 5XFAD mouse model has a significant sex difference in Aβ levels and amyloid deposition[17], we performed all statistical analyses for males and females separately. Four-month-old mice were sacrificed for this experimental group.

**Brain sample collection and sample process.** Mice were anesthetized with tribromoethanol (Avertin, 250 mg/kg, intraperitoneal injection, Sigma-Aldrich) and transcardially perfused with cold 0.1 M PBS, and then the brain was collected. For each mouse, one hemibrain was fixed in 4% paraformaldehyde overnight at 4 °C followed by: storage in 30% sucrose in PBS (pH 7.4) solution at 4 °C for the *Spi1*-knockdown study group, and storage in a 70% EtOH in PBS solution at 4 °C for the *Spi1*-overexpression study group. The other hemibrain was dissected, and cortical and hippocampal regions were flash-frozen, then stored at −80 °C until further analysis.

The cortical and hippocampal regions of the mouse brain were gently grinded with 0.1 M PBS in the presence of protease and phosphatase inhibitors (Roche) and then centrifuged at $20,000 \times g$ for 30 min at 4 °C and the supernatants (PBS fractions) were collected for further analysis. Next, we homogenized the remaining pellet with 1X RIPA lysis buffer (EMD Millipore) in the presence of protease and phosphatase inhibitors and then centrifuged at $20,000 \times g$ for 30 min at 4 °C. Supernatants (RIPA fractions) were transferred to fresh tubes, and the protein concentration was determined with a Pierce BCA Protein Assay Kit (Thermo Fisher Scientific). The remaining pellet was used for 5 M guanidine hydrochloride extraction (Guanidine fractions) using a rotator for 3 hrs at room temperature (RT).

**Western blot (WB) analysis.** RIPA fractions were used for the measurement of protein expression. Equal amounts of protein (15 μg) from each sample were loaded onto a 4−20% gradient SDS-PAGE gel (TGX gels, Bio-Rad Laboratories). Proteins separated by gel electrophoresis were transferred onto polyvinylidene difluoride membranes (PVDF; EMD Millipore) using an electrophoretic transfer system (Bio-Rad Laboratories), and the membranes were incubated 2 hrs at RT or overnight at 4 °C with the following primary antibodies: rabbit anti-β-Amyloid precursor protein (βAPP; 1:500; #51-2700; Invitrogen), rabbit anti-β-secretase 1 (BACE1; 1:1000; CS-5606s; Cell Signaling Technology) and mouse anti-β-actin (1:50,000; A1978, Sigma-Aldrich) as the loading control. After washing, the membranes were incubated with horseradish peroxidase-conjugated specific secondary antibodies (1:10,000) for 1 h at RT. The blots were developed with ECL WB detection reagents (GE Healthcare) and analyzed using a Luminescent Image Analyzer (Amersham Imager 680; GE healthcare).

**Measurement of Aβ concentrations.** The guanidine fractions were used for the quantitative determination of Aβ peptides using the V-PLEX Plus Aβ Peptide Panel 1 (6E10) Kit (K15200E; MESO Scale Discovery, MSD) according to the manufacturer's instructions. Briefly, plates were blocked with Blocker A (MSD) for 1 h at RT, and then the diluted guanidine fractions were added with SULFO-TAG-labeled anti-human Aβ 6E10 antibody and incubated for two hours at RT. Plates were shaken with an orbital shaker at 800 rpm in all incubation steps. After three additional wash steps, 1X Read buffer (MSD) was added to the plates. Signals were measured on a MESO QuickPlex SQ 120 (multiplexing imager, MSD).

**Histological analysis.** In the *Spi1*-knockdown study group, fixed hemibrains were frozen, and serial coronal sections (20 μm thickness) were obtained from rostral (bregma −1.22 mm) to caudal (bregma −2.70 mm) using a cryostat (CM 1860; LEICA Biosystems). Sections were stored in a cryoprotectant solution (50% glycerol in 0.1 M PBS) at −20 °C. Three sections spaced 460 μm apart were used for each

staining procedure. In the *Spi1*-overexpression group, fixed hemibrains were embedded in paraffin and sections (5 μm thickness) were obtained from rostral (bregma −1.58 mm) to caudal (bregma −2.18 mm) using a rotary microtome (LEICA Biosystems). Three sections spaced 200 μm apart were used for each staining procedure. After removal of paraffin in xylene and rehydration in a series of alcohol solutions (100%, 90%, and 70%), sections were steamed for 10 min in 10 mM sodium citrate buffer for antigen retrieval using a TintoRetriever Pressure Cooker (BSB 7008, Bio SB). To identify fibrillar plaques, sections were permeabilized with 0.25% Triton X-100 in PBS and stained with 10 μM X-34 in staining buffer (40% EtOH and 0.02 N NaOH in PBS). For immunohistochemistry, sections were treated with 10% MeOH and 3% $H_2O_2$ in PBS for 10 min to quench endogenous peroxidase and blocked with PBS containing 4% milk. Sections were then incubated with mouse anti-Aβ antibody (1:500, IBL10323; IBL-AMERICA) with 0.25% Triton X-100 in PBS containing 2% milk at 4 °C overnight. Sections were incubated with biotinylated anti-mouse antibody (BA-9200, Vector Laboratories) in blocking solution at RT for 1 h. Antibody binding was detected with Vectastain ABC Elite (PK6101; Vector Laboratories) and DAB development kits (SK-4100; Vector Laboratories) according to the manufacturer's instructions. Sections were dehydrated and mounted on slides with Permount (Fisher Scientific).

For immunofluorescence staining, sections were blocked with PBS containing 5% normal goat serum (NGS) or 5% normal donkey serum (NDS) at RT for 1 h, incubated with anti-IBA1 (1:1000 for cryosections and 1:400 for paraffin sections; ab178846; abcam), anti-GFAP (1:1000 both sections; Z0334; Dako Omnis), anti-Aβ (1:400, IBL10323; IBL), and anti-LAMP1 (1:200, ab24170; Abcam) antibodies in 2.5% NDS (or NGS) at 4 °C overnight, and then incubated with Alexa Fluor® 488 Donkey Anti-Rabbit (1:500, 712-545-152; Jackson Laboratory; for anti-IBA1, -GFAP, and -LAMP1) and Fluor® 568 Goat anti-Mouse (1:500, A-11011, Invitrogen; for anti-Aβ) antibodies at RT for 1.5 h. Sections were mounted on slides with Aqua mounting reagent (Polysciences, Inc.). All images were obtained with a digital pathology slide scanner (Aperio VERSA; LEICA Biosystems) or inverted fluorescence microscope (DMi8; Leica Biosystems).

**Image analyses.** For each mouse, three sections were used to obtain a single average data point per mouse. Aβ plaque levels, fibrillar plaque levels, microglial reactivity, astroglia reactivity, and axonal dystrophic levels were quantified using the Analyze Particles method within the NIH ImageJ program as we reported previously[11,48]. Images were converted from RGB color to 8-bit and then the threshold was set at 1.2% of the background level. The positive signal was determined by size, with a size larger than 2 pixels and less than 230 pixels (1 pixel = 6.45 μm) being identified as 82E1-positive (+) Aβ plaques, X-34+ fibrillar plaques, and LAMP1+ axonal dystrophy; a size larger than 8 pixels being identified as activated-IBA1+ microglia and GFAP+ astrocytes. Quantification data were expressed as the % of each positive signal load and the number of plaques/mm². These metrics were then compared between the genotypes.

The co-localization analysis of plaques and microglia was quantified by staining brain tissues with X-34 dye and IBA1 antibody. The quantification analysis was performed using CellProfiler software (Broad Institute of Harvard and MIT), colocalization metrics from a modular high-throughput image analysis[49]. Quantification data was normalized to the control group and expressed as the percentage of plaque area covered by IBA1-positive (+) microglia.

**In vitro Aβ uptake assay.** BV-2 microglia were seeded at a density of $3 \times 10^4$ cells/well in a 96-well plate. BV-2 microglial cells were transfected with *Spi1* siRNA and siRNA negative control (referred to as Control) or *Spi1* plasmid and its empty vector (referred to as Control) using Lipofectamin 3000 reagent (#L3000; Invitrogen). After 1day of

transfection, cells were incubated in opti-MEM (including N-2 supplement) medium containing Aβ labeled with pHrodo Red dye for 48 h. During incubation, the relative fluorescence unit (RFU) was measured on a microplate reader (Synergy H1, BioTek) at 1, 3, 6, 18, 24, and 48 h to determine the Aβ uptake level at Ex/Em 560/585 nm, respectively. After measuring the last RFU, we performed a CellTiter-Glo® 2.0 cell viability assay (#G9242; Promega) to compare the cell viability of each group. All experiments were performed $n = 5$ per group and repeated independently twice.

**RNA extraction and qRT-PCR.** Total RNA was extracted from mouse cortical tissues using TRI reagent® (Molecular Research Center, Inc) and quantified using a NanoDrop 2000 spectrophotometer (Thermo Scientific, Bremen, Germany). The extracted RNAs were reverse transcribed using the High-Capacity cDNA Reverse Transcription kit (Applied Biosystems) according to the manufacturer's instructions. Quantitative real-time PCR (qRT-PCR) was performed with FAST SYBR Green PCR Master Mix (Applied Biosystems) on a QuantStudio 3 (Applied Biosystems) using the default thermal cycling program. PCR amplification was performed using the specific primers listed in Supplementary Data 16.

**nCounter Mouse AD Panel analysis.** 75 ng of total RNA was hybridized for 16 h with capture probes and reporter probes from the AD-associated 770 genes nCounter Gene Expression code set. After hybridization, the probe-target complexes were scanned by a fluorescence microscope and labeled barcodes were counted. Gene expression analysis was performed using the nCounter system (NanoString Technologies) according to the manufacturer's instructions. Following the manufacturer's guidelines, data were analyzed using the nSolver Advanced analysis software (NanoString Technologies) with built-in quality control, normalization, and statistical analyses. Data were exported from nSolver and imported into R (v3.6.2) in RStudio (v1.2.5033). LIMMA (v3.40.6) was used to fit linear models of counts ~ Genotypes for WT (*Spi1*^+/+^;APP/PS1) versus KD (*Spi1*^+/−^;APP/PS1) and WT (*Spi1*^+/+^;5XFAD) versus TG (*Spi1*^Tg/0^;5XFAD) comparisons. Pathway analysis was performed by EnrichR (WikiPathways 2019 Mouse) and MetaCore software; GO processes, Process Networks, and Transcription factor analyses were performed using MetaCore software. A protein-protein interaction network analysis was performed using the STRING database (v11.0). Network analysis was set at medium confidence (STRING score = 0.4) and the disconnected nodes in the network were not considered.

**Single-cell RNA library preparation.** Four-month-old *Spi1*^+/+^;5XFAD and *Spi1*^Tg/0^;5XFAD mice were anesthetized and perfused with cold PBS, and then the brain was collected. After removing the cerebellum, the right hemisphere was immediately finely minced on ice and transferred to a polypropylene tube containing cold Accutase (A11105-01; Gibco). Tissue was incubated at 4 °C for 30 min and then centrifuged at 300 × *g* for 5 min. The supernatant was aspirated and the samples were resuspended with cold wash buffer (HBSS containing 0.04% BSA), and gently dissociated through trituration. The dissociated cell suspension was passed through 70 μm and 40 μm cell-strainers sequentially, then centrifuged at 300 × *g* for 10 min. The supernatant was aspirated, and cells were resuspended in myelin removal buffer (DPBS containing 0.5% BSA). Myelin removal beads (Miltenyi) were added to each sample and mixed thoroughly. Samples and beads were incubated for 15 min at 4 °C with rotation, washed with myelin removal buffer, and then centrifuged at 300 × *g* for 10 min. The supernatant was aspirated, and samples were resuspended in myelin removal buffer. LS columns (130-042-401; Miltenyi) were loaded into a QuadroMACS separator on a MACS MultiStand with Tube Rack (130-090-976, 130-042-303, and 130-091-052; Miltenyi). Columns were

equilibrated with myelin removal buffer. Cell suspensions were loaded onto columns, and myelin-depleted cell suspensions were washed through using myelin removal buffer. Cells were pelleted through centrifugation at 300 × *g* for 5 min. The supernatant was aspirated and cells were gently resuspended with resuspension buffer (DPBS containing 0.04% BSA). Cell numbers and viability were quantified using trypan blue staining and observed with an EVOS XL Core microscope. Single-cell suspensions were prepared from the brains of two *Spi1*^+/+^;5XFAD and two *Spi1*^Tg/0^;5XFAD mice, yielding 16,532 cells from *Spi1*^+/+^;5XFAD mice and 18,247 cells from *Spi1*^Tg/0^;5XFAD mice. These suspensions were processed in a single batch by the 10X Chromium. Each single cell mix was loaded into a Chip G and run on the Chromium Controller for GEM generation and barcoding. Sample processing and library preparation were performed according to the manufacturer's instructions using the Chromium Next GEM Single Cell 3' v3.1 dual index kit (10X Genomics) and SPRIselect paramagnetic bead-based chemistry (Beckman Coulter Life Sciences). The cDNA and library quality were assessed using the 2100 Bioanalyzer and a High Sensitivity DNA kit (Agilent Technologies). The final library concentration was determined using a QuBit Fluorometer and the dsDNA HS assay kit (Thermo Fisher Scientific). Sequencing was carried out on a NovaSeq 6000 (v1.5 S2; Illumina) with 28-10-10-91 read setup.

**Single-cell RNA sequencing data analysis.** Processing of the sequencing data was performed with the *cellranger* pipeline (v4.0.0, 10X Genomics). The filtered feature-cell barcode matrices (including the hashtag count matrix) generated by CellRanger were loaded into SoupX (v1.4.8)[50] in Rstudio (v1.4.1717) running R (v4.0) and cleaned using default settings. Cleaned data were then loaded into Seurat (v4.0.1)[51]. During Quality Control cells with >10% mitochondrial reads, or unique reads below 200 or above 3212 (median +1.5 times standard deviation) were excluded from analysis. Data were log-normalized, and cells were clustered using the first 18 Principal Components based on an Elbow Plot. Cluster marker genes were identified using the FindConservedMarkers function. We used the scMCA function (v0.2.0) provided by the Mouse Cell Atlas[29], to annotate cluster identity. Differences in the proportion of cells per cluster between genotypes were tested using a Two-way ANOVA with *Sidak's* multiple comparisons test in GraphPad Prism (v9.1.0, GraphPad Software). GO and KEGG analyses were performed on differentially expressed genes within microglia using Gprofiler2 (v0.2.0). Single Cell Weighted Gene Correlation Network Analysis was performed using scWGCNA (v0.1.0)[31], after merging *Spi1*^+/+^;5XFAD and *Spi1*^Tg/0^;5XFAD samples into 2 separate Seurat objects. The network plot of the light cyan module was generated using Cytoscape (v3.8.0)[52]. CellChat (v1.1.2)[32] was used to infer inter-cluster signaling using default package settings.

**Statistics.** Several statistical tests were used throughout, as mentioned in each relevant section. All data are expressed as mean ± SEM. The mean values from each mouse were used to compute the statistical differences. Individual data points are shown when possible. Statistical analyses were performed with GraphPad Prism (v9.1.0), Metacore algorithms, and R (v3.6.2) in RStudio (v1.2.5033).

The statistical analyses were performed by comparing the means of different groups using unpaired two-tailed *t*-tests, two-tailed *Fisher's* exact test, two-way ANOVA with *Sidak's* multiple comparisons test, and two-way ANOVA with *Ficher's* LSD multiple comparisons test. The false discovery rate (FDR) was calculated using the *Benjamini–Hochberg* method. $P < 0.05$ was considered statistically significant.

**Reporting summary**
Further information on research design is available in the Nature Portfolio Reporting Summary linked to this article.

## Data availability

The scRNA-seq data generated in this study have been deposited in the GEO database (GSE222624). The NanoString data (GSE225669) (Supplementary Data 2 and 5) was included in the Supplementary Information of this paper. The biological functions enrichment data generated in this study are provided in the Supplementary Data and Source Data file.

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

## Acknowledgements

We thank MODEL-AD and Biomarker Core at Stark Neurosciences Research Institute for sharing the nCounter Analysis System (NanoString), MESO QuickPlex SQ120 (multiplexing imager) instrument, and the LEICA Aperio VERSA (digital pathology slide scanner). Paraffin-embedding and sectioning services were provided by the Histology and Histomorphometry Core at the Indiana University Musculoskeletal Health Center. Single-Cell RNA libraries were sequenced at the Center for Medical Genomics at Indiana University School of Medicine, which is partially supported by the Indiana Genomic Initiative at Indiana University (INGEN); INGEN is supported in part by the Lilly Endowment, Inc. We thank Drs. S. Louise Pay and Younghye Moon for manuscript editing. This study was supported by funding from Indiana University (Strategic Research Initiative fund, Precision Health Initiative fund, and P. Michael Conneally Professorship) and NIH R01AG077829, R01AG071281, R21AG072738, and RF1AG074543 (J.K.). B.K. was supported by Alzheimer's Association Fellowship (AARF-21-852175). L.C.D. and S.P. were supported by Eli Lilly-Stark Neuroscience Fellowship. M.D.T. was supported by an NIH F31 (F31AG074673). H.K was supported by Sarah Roush Memorial Fellowship in Alzheimer's Disease. M.M.A-A. was supported by an NIH T32 (T32AG071444). H.R.S.W. was supported by an NIH F30 (F30AG079580).

## Author contributions

B.K. and J.K. conceived and design of the project. B.K., L.C.D., M.D.T., H.K., A.D.S. and J.H.P performed the experiments. B.K., L.C.D., M.D.T., H.K., A.D.S., D.J.A., H.R.S.W. and M.M.A-A. analyzed the data. B.K., L.C.D., M.D.T, H.K. and J.K. wrote the paper. All authors reviewed the manuscript.

## Competing interests

The authors declare no competing interests.
