## [Peer Review File · Nature Communications]

Effects of SPI1-mediated transcriptome remodeling on Alzheimer's disease-related phenotypes in mouse models of A β amyloidosisReviewers' Comments:

Reviewer #1:

Remarks to the Author:

This is an important and timely manuscript looking at PU.1 transcription factor and how it impacts AD. This gene (SPI1) was recently implicated as an AD gene and although we know it to be a "microglial" gene, we have not yet determined what it is doing functionally in the context of AD pathogenesis. This manuscript uses separate knockdown or overexpression models of PU.1 and two separate AD models (though see below). This loss and gain approach, combined with the adequate sample sizes allows for strong conclusions to be drawn. Though clearly more work needs done in other AD models (e.g. tau), this seminal paper will be the first to clearly demonstrate what it is doing in the context of amyloid at least. There are few concerns, mostly methodological, that require clarification before publishing this manuscript.

Please provide a little more description of the SPI1 knockdown mice before launching into text as to how they were used (i.e. how the allele was created and characterized, why it's not a full KO, etc).

Ext Fig 1. Please indicate whether this is all males, females, or both in the legend and how many of each sex.

Language is a bit much as the effects are significant, but not "dramatic" or "marked"

Add data regarding ratio of fibrillar to total plaque load (e.g. X34/82E1).

Add data regarding CAA if available.

"Moreover, there was a significant correlation between A β 40 and A β 42 levels in both cortex and hippocampus of Spi1+/-;APP/PS1 and Spi1+/+ ;APP/PS1 mice (fig. S2, E and F), demonstrating the rigor of our biochemical analysis." Remove this odd sentence as it does nothing to demonstrate the rigor of the biochemical analysis. Also remove the similar sentence on page 8, lines 170-171.

Clarify in the text whether nanostring and proteomic data in Fig 2 is from bulk tissue. It seems that it is, but please state this in the text.

Proteomic data does not make sense. First, SPI1 is actually INCREASED in the KD vs control mice, how could this be? Second, and more importantly, there is no formal analysis of the data for significantly differentially expressed proteins, just a list of the levels in N=3 v N=3. This suggests that there were, in fact, zero DEP's and the WGCNA is just an attempt to glean something useful from largely negative data. If there were no significant DEP's, the proteomic data should be removed from the manuscript entirely. It's not necessary, and the claims made based on this data are unfounded.

What is the rationale for switching between the Jucker model and the 5xFAD model?

The scRNAseq analysis is missing some important quality control figures. How many mice from each group were used? What is the purity of the clusters? E.g. Camk2a is a neuronal gene and likely means there was ambient RNA issue or that the clusters were not sufficiently cleaned. What are the relative cell proportions from each genotype in each of the clusters? What are the "DAM" clusters of microglia?

The scRNAseq is also presented in an odd way. Please enlarge the UMAPs and write out the annotations directly on the figure rather than (N1, En, A1, Er, etc) as those are not intuitive. Also, add a few gene names to each of the volcanoes shown in Ext Fig 6.

There is greater IBA1 and GFAP load in SPI1-KD mice and is less in the SPI1-Tg mice, but is that just because they have more or less plaque load? Same for LAMP1 staining. The authors should either normalize these changes to plaque load, or (better) do a localized analysis (e.g. Sholl). Either answer is fine, but worth knowing this data.

Reviewer #2:

1 **Review for Nature Communications. NCOMMS-23-30155-T**

2

3

4 **Comments to the Author:**

5 In this manuscript, authors utilized 5xFAD crossed SPI1-knockdown and transgenic model
6 mice to confirm the controversial role of *Spi1* in Alzheimer's disease (AD). It is interesting that
7 microglial inducing transcription factor PU.1(*Spi1*) shows protective effect on A β amyloidosis.
8 Consistent *in vivo* results support the protective function of *Spi1* in Alzheimer's disease. Multi-
9 omics approach to analyze the diverse effects of *Spi1* and the usage of both male and female
10 mice to check gender difference enhances the quality of the study. However, the interpretation
11 and discussion appear to be weak compared to the various analyses conducted in the study. It
12 would be desirable to have additional biochemical data to corroborate the results of
13 transcriptomics and proteomics data. Therefore, some experimental results remain to be
14 clarified or improved.

15

16

17 Major comments

18

1. Authors experimentally show that overexpression of *Spi1* protects 5xFAD mice from
19 amyloidosis and knockdown worsens APP/PS1 mice. it would be helpful to address
20 whether the genetic variant of *SPI1* in GWAS study was a loss of function or gain of
21 function and relate it to the results of the *in vivo* knockdown and transgenic model
22 experiments in the paper.

23

2. In Fig.4d-i, could authors describe any differences in microglia clusters or cell-cell
24 communications between NTG and TG mice? The main theme of the study would be
25 the effect of *SPI1* on cellular response. The organization of the figure seems to be more
26 focused on extracting microglia cluster (in here, M4, 8, and 10) that enriched in DAM-
27 like gene network or cell-cell communication between clusters.

28

3. In lane 129-133, what is an interpretation for upregulated royal-blue modules such as
29 lipid metabolism, neuronal development and gamma-secretase proteolysis?
30 Transcription factor *Spi1* would primarily affect microglia or other myeloid cells.
31 Unlike the result from proteomics data, authors checked beta-secretase expression and
32 end products in extended Fig.5. What is the author's opinion on which cells' gamma-

33 secretase function in brain tissue would be regulated by changes in *Spi1* gene
34 expression and how?

35 4. From integrative transcriptomics analysis, authors extracted biological pathways such
36 as cytokine, complement, immune responses (Fig.2, Fig.5). *Clq*, *Trem2*, *Tyrobp*, *Syk*
37 genes are closely related each other. Their protein products have an important role in
38 phagocytosis and microglia clustering. Did proteomics data also support their
39 expression data which regulated by PU.1.?

40 5. In line with previous questions, the study lacks a biochemical data about A β clearance.
41 Authors showed that *Spi1*-overexpression decreases A β burden (Fig1, Fig.3, and
42 extended Fig.5). As authors confirmed that *Spi1* gene does not affect APP processing
43 (extended Fig.3), it would be worth to check how did *Spi1* overexpression reduced A β
44 burden and whether glial A β clearance function have changed.

45 6. In lanes 262-266, There are some doubts about the interpretation. "Additionally, we
46 observed signaling from a small population of T cells to microglia through the same
47 pathway", which suggests that T cells may also be involved in the amyloid pathology
48 model. It is a surprise that small portions of T cells have observed in sequencing
49 analysis (Fig. 4d). However, a recent article that referenced in this paper compares
50 APOE4;APP/PS1 or APOE4;5xFAD and APOE4/TauP301S mouse models and
51 provides data showing that amyloidosis model has very small numbers of T cells in
52 parenchyma which comparable with APOE4;WT mice, but tauopathy model has
53 significantly increased numbers of T cells. The literatures used to make curation in
54 these types of analyses will also include studies performed in peripheral tissues.
55 Therefore, it can be misleading to make claims based on analytics data alone. Since
56 this is far from the main issue of the paper, authors can simply tone down the discussion.
57 If the authors want to make a point, they should perform some biological validation,
58 such as staining, as shown in the references.

59

60 Minor comments

61 1. The group names in the figure and the manuscript don't match well. For example,
62 abbreviation WT means *Spi1*^{+/+};APP/PS1 mice. I understand that it's a wild type for
63 *Spi1*, but it can be confusing. Then NTG should be displayed as WT as well, or just

64 describe them both as Spi^{+/+};5xFAD (extended Fig.3). Authors should use official
65 group names as they used in the manuscript or use more appropriate abbreviations.

66 2. In Fig.4e and extended Fig.8, it is curious that microglia cluster M4,8 and 10 are
67 enriched with DAM genes but Fig.4e lacks M10 cluster information. Some of the same
68 genes repetitively observed in multiple modules in chord plot (Fig.4h). Are some
69 clusters significantly different enough to be categorized separately?

70 3. “Effects of~” term is somewhat neutral to be used in the title. Authors should better
71 select other words to write a title that captures the conclusion of the study.

72

73

74

Reviewer #3:

Remarks to the Author:

Recent work has implicated Spi1, which encodes PU.1, as a genetic risk factor for Alzheimer's disease (AD). Spi1 is primarily expressed by microglia in the brain and there has been a few studies which have elucidated how it might play a role in AD. However, these studies have been in vitro and to my knowledge there has not been an in vivo study modulating the levels of Spi1 in an amyloid model. In this manuscript, Kim et al attempts to address this gap in the literature. First, they crossed APP/PS1-21 mice to Spi1^{-/-} to obtain Spi1^{-/+}; APP/PS1-21 (KD) or Spi1^{+/+};APP/PS1-21 (WT) mice in order to understand the effect of knocking down Spi1 on amyloidosis. They found that KD mice had significantly more amyloid than WT mice by several methodologies. They then performed Nanostring gene analysis, proteomics, and subsequent pathway analyses to determine by what mechanism Spi1 modulation impacts amyloidosis. According to their analysis, they determined that Spi1 KD resulted in a reduction in microglial phagocytosis, and this was potentially responsible for the increase in amyloidosis. Next, they crossed 5XFAD mice to Spi1Tg/0 to generate either Spi1Tg/0; 5XFAD (TG) or 5XFAD with non-transgenic Spi1 (NTG) to investigate the effect of overexpression of Spi1 on amyloidosis. Consistent with their previous experiment, they found that overexpression of Spi1 lowered amyloidosis. They then performed both Nanostring gene analysis and single-cell RNA sequencing (scRNAseq) comparing NTG and TG mouse cortices. According to their Nanostring analysis, the microglial phagocytosis pathway was surprisingly decreased in TG mice compared to NTG mice. In their scRNAseq experiment, they captured 16,456 cells that primarily contained microglia. They were able to identify 19 clusters, 11 of which were microglia. They found that cluster 4 was particularly enriched in DAM markers. Using the CellChat program, they found that microglial cluster 4 may signal to other microglial clusters through Ccl3 and Ccl4 by the Ccr5 receptor. In looking at the DEGs from the KD and overexpression experiments together, they found that 6 genes change transcriptionally in opposite directions. Lastly, they found that Spi1 overexpression reduces gliosis and decreases dystrophic neurites.

Although this study addresses an important question and the amyloid, gliosis, and dystrophic neurite quantifications are sound, all meeting the standards of Nat. Comm., there seems to be several methodological problems in the multi-omic analyses that limit the ability to draw conclusions into the mechanism whereby Spi1 modulation alters amyloidosis.

Major comments

- For the KD vs WT Nanostring experiment in Figure 2, none of the "DEGs" p-adj are <0.05 (Supplementary table 2). All the genes called "DEGs" are based on the raw p-values, which are not corrected for multiple testing (FDR adjusted p-values). It is unclear whether all these DEGs might just be false positives. If the authors believe these changes aren't false positives, they should prove it using another method such as qPCR. Also, this caveat should be explicitly stated in the paper, so readers can make their own conclusions about the data without assuming that the DEGs are derived from FDR corrected p-values.
- Most of the pathways that are "enriched" in Figure 2B-C have 1-2 genes out of the pathway (Table 3) and due to the issue addressed above it is arguable whether these genes have altered expression when Spi1 is KD.
- The proteomics data in Figure 2 is hard to interpret. There is no volcano plot showing differentially expressed proteins. As far as I can tell, there are no stats comparing proteins found between conditions in the supplementary table 5-6. There is only the WGCNA analysis which is difficult to interpret without knowing which proteins were changed by Spi1 KD. The authors do go on to identify proteins that are in the royal blue enriched module, but it is unclear whether these proteins were actually changed significantly by Spi1 KD or they are just in the overall module which was changed. In other words, the authors focus on changes in WGCNA modules but do not disclose which proteins in these modules were changed by Spi1 KD.
- For the TG vs NTG Nanostring experiment in Figure 4, only 2 of the "DEGs" p-adj are <0.05 (Supplementary table 12). As mentioned above, the genes with a raw p-value <0.05 but not with a FDR corrected p-value <0.05 might just be false positives. If the authors believe these changes aren't

false positives, they should prove it using another method such as qPCR. Also, this caveat should be explicitly stated in the paper, so readers can make their own conclusions about the data without assuming that the DEGs are derived from FDR corrected p-values.

- For the single cell experiment, it should be explicitly stated how many biological replicates the data is derived from and how many cells came from which animal. I could not find that information and without it the data from the experiment is hard to interpret.
- The analysis of the single cell experiment seems to be mostly limited to examining microglial markers in the integrated dataset (containing TG and NTG cells) rather than making any comparisons between the two. There are no proportion differences in clusters shown between the TG and NTG groups. Supplementary table S15 is labeled "Cell-type clusters DEGs between Spi1Tg/0;5XFAD and 5XFAD mice." I believe that this is actually the DEGs (which are FDR corrected) between all the clusters for the integrated dataset rather than a comparison of DEGs between TG and NTG for each cluster. This seems to correspond with extended data figure 6. Please make this clearer if this interpretation is wrong and these are actually differentially expressed genes between the TG and NTG conditions. Without direct comparisons between the TG and NTG groups, it is unclear what the reader should derive from this information that addresses the scientific question posed about the relationship between Spi1 overexpression and amyloidosis.
- Without knowing the expression levels of Ccl3/4 and Ccr5 in the NTG and TG conditions, it is unclear how the CellChat findings in Figure 4 are relevant to the scientific question at hand.

Minor comments

- It would be interesting to provide data on plaque size for the KD and overexpression studies since that could be easily calculated. Do the changes to microglia affect only plaque number or also plaque size?
- Based on the proteomic WGCNA analysis, there seems to be little overlap with the Nanostring analysis. Although this is not surprising given the accumulating studies that show poor correlation between transcriptomics and proteomics, this is not acknowledged in the paper and the authors should try to speculate as to the reason for this.
- Different amyloid mouse models used for KD (APP/PS1-21) and overexpression (5XFAD) experiments may limit the comparison of the experiments and could be the reason only 6 genes seemed to go in opposite directions from the KD and overexpression datasets.

Reply to the reviewer's comments:

We thank all reviewers for their insightful feedback and helpful critiques of our manuscript. Based on their constructive inputs, we now have thoroughly addressed all comments, significantly improving the quality of our manuscript. All major changes are marked with track changes in the manuscript for your convenience (in the Microsoft Word file, not in the merged PDF file).

Reviewer #1 (Remarks to the Author):

This is an important and timely manuscript looking at PU.1 transcription factor and how it impacts AD. This gene (SPI1) was recently implicated as an AD gene and although we know it to be a “microglial” gene, we have not yet determined what it is doing functionally in the context of AD pathogenesis. This manuscript uses separate knockdown or overexpression models of PU.1 and two separate AD models (though see below). This loss and gain approach, combined with the adequate sample sizes allows for strong conclusions to be drawn. Though clearly more work needs done in other AD models (e.g. tau), this seminal paper will be the first to clearly demonstrate what it is doing in the context of amyloid at least. There are few concerns, mostly methodological, that require clarification before publishing this manuscript.

1. Please provide a little more description of the SPI1 knockdown mice before launching into text as to how they were used (i.e. how the allele was created and characterized, why it's not a full KO, etc).

Response: We appreciate this suggestion that will help readers better understand this mouse model. To make *Spi1* knockdown mice, the -14 kb upstream regulatory element (URE) of the *Spi1* was deleted, resulting in reduced PU.1 expression¹. Since our *Spi1* Knockdown mouse is publicly available from Jackson Laboratory, we provided the catalog number and cited the original research article describing the creation of this mouse model in the Methods section. Additionally, we now further elaborated on why we chose this mouse line over the conventional *Spi1* whole-body knockout mouse.

- Page 4, line 63-68: In the result section, we now added “The conventional *Spi1*-knockout mice have early lethality phenotype². Therefore, we decided to utilize a *Spi1*-knockdown mouse model in which the -14 kb upstream regulatory element of the *Spi1* was deleted¹. Because homozygous *Spi1*-knockdown mice also become moribund from T-cell lymphoma and acute myeloid leukemia starting at 3 months of age¹, we utilized only *Spi1* heterozygous knockdown (*Spi1*^{+/-};APP/PS1) mice and wild-type for *Spi1* (*Spi1*^{+/+};APP/PS1) mice.”

2. Ext Fig 1. Please indicate whether this is all males, females, or both in the legend and how many of each sex.

Response: Only male *Spi1*-mutant mice and their littermate-control mice were used for experiments in Extended Figure 1. We now modified the legend of Extended Figure 1 accordingly.

3. Language is a bit much as the effects are significant, but not “dramatic” or “marked”

Response: In response to the reviewer's suggestions, we have replaced terms like "dramatic" or "marked" with "significant" to convey the magnitude of the observed changes more accurately.

4. Add data regarding ratio of fibrillar to total plaque load (e.g. X34/82E1).

Response: In accordance with the reviewer's suggestion, we now analyzed the ratio of fibrillar to total plaque load. The ratio of fibrillar plaque to the total plaque load remained unchanged for both mouse models. This finding suggests that the effect of *Spi1* level extends beyond a specific plaque form, impacting the overall amyloid plaque load. We now added the following data to Extended Data Figures **2h,i** and **4f,g** and updated the main manuscript.

Updated Extended Data Fig. **2h,i** (left panel) and **4f,g** (right panel)

- Page 5-6, line 101-106: In the result section, we now added "To determine whether *Spi1* knockdown affects only certain form of amyloid, such as fibrillar versus diffuse plaques, we analyzed the ratio of fibrillar plaque to total plaque load. No difference in that ratio was observed (Extended Data Fig. **2h,i**). This finding suggests that *Spi1* knockdown did not have any preferential effect on fibrillar or diffuse plaques, but rather increased the overall amyloid plaque load. "

- Page 8, line 161-163: In the result section, we added "Consistent with the effect of *Spi1*-knockdown on plaques (Extended Data Fig. **2h,i**), the ratio of fibrillar plaque to total plaque load remained unchanged upon *Spi1*-overexpression (Extended Data Fig. **4f,g**)."

5. Add data regarding CAA if available.

Response: The two mouse models we employed unfortunately do not develop reliable CAA pathology within the age range we examined. To be able to assess the effect of *Spi1* on CAA pathology, we need mouse models that develop more robust CAA pathology.

6. "Moreover, there was a significant correlation between Aβ40 and Aβ42 levels in both cortex and hippocampus of *Spi1*^{+/-};*APP/PS1* and *Spi1*^{+/+};*APP/PS1* mice (fig. S2, E and F), demonstrating the rigor of our biochemical analysis." Remove this odd sentence as it does nothing to demonstrate the rigor of the biochemical analysis. Also remove the similar sentence on page 8, lines 170-171.

Response: To address the reviewer's critiques, we now removed the reference to the rigor of the biochemical analysis.

7. Clarify in the text whether nanostring and proteomic data in Fig 2 is from bulk tissue. It seems that it is, but please state this in the text.

Response: We now clarified by stating in the text that the NanoString data was from cortex tissue. Following the reviewer's suggestion, we now removed our Proteomics data, hence this issue is no longer considered.

8. Proteomic data does not make sense. First, SPI1 is actually INCREASED in the KD vs control mice, how could this be? Second, and more importantly, there is no formal analysis of the data for significantly differentially expressed proteins, just a list of the levels in N=3 v N=3. This suggests that there were, in fact, zero DEP's and the

WGCNA is just an attempt to glean something useful from largely negative data. If there were no significant DEP's, the proteomic data should be removed from the manuscript entirely. It's not necessary, and the claims made based on this data are unfounded.

Response: We acknowledge the reviewer's concerns regarding the proteomics data.

8-1) Apparent increase in SPI1 level in SPI1 Knockdown mouse versus wild-type control mice

We now plotted the SPI1 protein levels based on proteomics data below. There was no significant difference in *Spi1* protein levels in the cortex of four-month-old *Spi1*-wildtype (WT) and *Spi1*-knockdown (KD) mice **on APP/PS1 background**.

We understand that no reduction in SPI1 protein level in the SPI1 Knockdown mouse model might look counterintuitive. We believe this happened because amyloid pathology may increase *Spi1* expression levels. This hypothesis aligns well with a previous study that reported elevated *Spi1* expression in APP/PS1-tg mice compared to littermate WT mice at four months of age³, using the same mouse model employed in our study. In addition, *SPI1* expression was increased in the frontal gyrus of postmortem brain tissues with AD compared to age-matched controls⁴. This supportive evidence further strengthens our hypothesis, given the exacerbated amyloid pathology observed in our *Spi1* KD data set.

To directly address the critical question, we now measured the levels of SPI1 mRNAs and proteins in mice without amyloid pathology. As demonstrated in our updated Extended Fig. 1, *Spi1* KD mice exhibit significantly reduced SPI1 mRNA (a) and protein (b) levels at two months of age as demonstrated in qPCR and WB analyses, respectively. In addition, we now added the SPI1 mRNA (c) and protein (d) levels from *Spi1* TG mice as well in the Expanded Data Fig. 1.

Updated Extended Data Fig. 1

8-2) Regarding "the proteomic data should be removed from the manuscript entirely. It's not necessary."

Following the reviewer's suggestion, we now remove all data related to proteomics.

9. What is the rationale for switching between the Jucker model and the 5xFAD model?

Response: Our transition from APP/PS1-21 amyloid mouse model to the 5xFAD amyloid mouse model was mainly due to an administrative/logistical issue, not a scientific concern. Specifically, we had previously secured an MTA agreement for APP/PS1-21 mouse model when our lab was located at the Mayo Clinic, Jacksonville, FL for the SPI1 knockdown mouse project. However, due to the absence of an MTA agreement at Indiana University where our lab was later relocated, we opted to generate PU.1 OE mice on the 5xFAD background.

We thought using two different APP mouse models would even increase the rigor of our study. In fact, our findings demonstrate that *Spi1*-mediated effects are not limited to only one particular APP mouse model. We now believe obtaining consistent data across two different APP mouse models is one of the main strengths of our manuscript.

10. The scRNAseq analysis is missing some important quality control figures. How many mice from each group were used? What is the purity of the clusters? E.g. *Camk2a* is a neuronal gene and likely means there was ambient RNA issue or that the clusters were not sufficiently cleaned. What are the relative cell proportions from each genotype in each of the clusters? What are the "DAM" clusters of microglia?

Response: The reviewers have raised several key questions about our single-cell RNA-sequencing experiment which were not fully explained in the original submission. We are grateful for these suggestions on how to increase the clarity of our manuscript.

- Page 37, line 889-892: In the methods section, we now added "Single-cell suspensions were prepared from the brains of two *Spi1*^{+/+};5XFAD and two *Spi1*^{Tg/0};5XFAD mice, yielding 16,532 cells from *Spi1*^{+/+};5XFAD mice and 18,247 cells from *Spi1*^{Tg/0};5XFAD mice. These suspensions were processed in a single batch by the 10X Chromium."

In addition, Reviewer 1 raises a key point about cluster purity. Ambient RNA is a frequent confounder of many single-cell RNA-seq data. We routinely use the package SoupX to infer and remove signals originating from ambient RNA (line 905; in the method section). Rho values (estimates of the fraction of contaminating reads) are low, being 0.1, and 0.02 for *Spi1*^{+/+};5XFAD mice, and 0.13 and 0.01 for the *Spi1*^{Tg/0};5XFAD mice.

The specific gene mentioned, *Camk2a*, does appear as a differentially expressed gene in non-neuronal clusters (e.g. Table **S12**. Cell-type clusters DEGs between *Spi1*^{Tg/0};5XFAD and *Spi1*^{+/+};5XFAD mice - Microglial Cluster 1, Log₂FC 0.59, Adjusted P 4.07 x10⁻⁹). We plotted the normalized RNA expression of *Camk2a* and found that while its expression was very high in Astrocyte 1 and Neuron 1, it was also detected in microglial clusters at lower level, including Microglial Clusters 1, 5, and 7. This figure is shown below.

Supporting our findings, a previous study demonstrated the expression of CamK2a in microglia⁵.

Furthermore, as indicated by the single-cell RNA sequencing data from the Mouse Cell Atlas website (<https://bis.zju.edu.cn/MCA/search2.html>) published in Han, X., et al., Mapping the Mouse Cell Atlas by Microwell-Seq. *Cell*, 2018. 172(5): p. 1091-1107 e17., the expression of the CamK2a gene is most prominent in neurons, but it is not exclusive to neurons. The expression levels of Cam2ka in this public dataset are shown below.

Therefore, the detection of Camk2a transcripts in microglia does not seem to be an artifact of ambient RNA, as otherwise it should have been removed from these cells by SoupX.

To address the reviewer’s question about relative cell proportions from each genotype within each cluster, we employed Propellor, a Bayesian tool for analyzing changes in cell proportions per cluster per condition. While we observed variations in proportions between cells per cluster per genotype, none of these differences reached statistical significance (p -values ranging from 0.973 to 0.928). We now added the figure below in the Expanded Data Fig. 7 and stated that “There were no significant differences in cell populations within each cluster between the two genotypes (Extended Data Fig. 7 and lines 291-292)”.

Ratio of average cell proportions per cluster per genotype. Shown are bar plots per each cell cluster (M1-11=microglia; A1-2=astrocytes; En=endothelial cells; Er=erythrocytes; Ma=macrophages; N1=neurons; O=oligodendrocytes; and T=T-cells) representing the ratio of average cell proportions between the genotypes (Cells in *Spi1^{Tg/0}*;5XFAD mice/Cells in *Spi1^{+/+}*;5XFAD mice). *Benjamini-Hochberg*-adjusted *P*-values are shown at the end of each bar. *P*-values are shown at the end of each bar.

To address the reviewer’s enquiry about the **Disease Associated Microglia (DAM)**, we analyzed microglia clusters in our dataset. We identified the enrichment of DAM signature genes and reduction of homeostatic microglial genes in clusters 4, 8, and 10 (Extended Data Fig. 10a and lines 309-313). We focused on microglial cluster 4, the cluster with the highest DAM expression to increase the robustness of our analysis during the CellChat analysis (line 325-328). For the reviewers’ convenience, we have attached the Extended Data Figure 10a below.

11. The scRNAseq is also presented in an odd way. Please enlarge the UMAPs and write out the annotations directly on the figure rather than (N1, En, A1, Er, etc) as those are not intuitive. Also, add a few gene names to each of the volcanoes shown in Ext Fig 6.

Response: We now made the requested changes as shown below.

Updated Fig 9a

Updated Extended Data Fig. 8 (previously Ext Fig. 6)

12. There is greater IBA1 and GFAP load in SPI1-KD mice and is less in the SPI1-Tg mice, but is that just because they have more or less plaque load? Same for LAMP1 staining. The authors should either normalize these changes to plaque load, or (better) do a localized analysis (e.g. Sholl). Either answer is fine, but worth knowing this data.

Response: We now normalized the load of IBA1, GFAP, and LAMP1 to the total plaque load. The ratios of IBA1,

GFAP, and LAMP1 to the total plaque load remained unchanged in both mouse models, suggesting that their alterations might be driven by the plaque load.

Not only the total area covered by microglia but also microglia clustering around the plaques is important given their role in A β uptake. We agree with the reviewer that a more careful analysis of the relationship between microglia and plaques will be a better approach. Therefore, we now performed a co-localization analysis of plaques and microglia to investigate whether the *Spi1* level could affect microglia coverage of plaques. Our findings now revealed a significant reduction in plaque-associated microglia in *Spi1* KD mice compared to WT mice (Fig. 7a,b), while *Spi1* TG mice showed a trend of increase in plaque-associated microglia relative to *Spi1* WT mice (Fig. 7c,d). These findings suggest that *Spi1* level, not merely plaque load, impacts microglia functions.

Most importantly, to further explore this relationship, we now conducted fibrillar A β (fA β) uptake assay to determine whether *Spi1* expression levels could modulate microglial A β clearance (Fig. 7e,f). BV-2 microglial cells were transfected with a *Spi1* siRNA or plasmid to knockdown or overexpression of *Spi1*, respectively. Twenty-four hours post-transfection, we conducted a time-course analysis of A β uptake using pHrodo-labeled fA β . As demonstrated below, *Spi1*-knockdown decreased fA β uptake (Fig. 7e), while *Spi1*-overexpression increased it (Fig. 7f).

Updated Fig. 7

Reviewer #2 (Remarks to the Author):

In this manuscript, authors utilized 5xFAD crossed SPI1-knockdown and transgenic model mice to confirm the controversial role of *Spi1* in Alzheimer's disease (AD). It is interesting that microglial inducing transcription factor PU.1(*Spi1*) shows protective effect on Aβ amyloidosis.

Consistent *in vivo* results support the protective function of *Spi1* in Alzheimer's disease. Multi omics approach to analyze the diverse effects of *Spi1* and the usage of both male and female mice to check gender difference enhances the quality of the study. However, the interpretation and discussion appear to be weak compared to the various analyses conducted in the study. It would be desirable to have additional biochemical data to corroborate the results of transcriptomics and proteomics data. Therefore, some experimental results remain to be clarified or improved.

Major comments

1. Authors experimentally show that overexpression of *Spi1* protects 5xFAD mice from amyloidosis and knockdown worsens APP/PS1 mice. It would be helpful to address whether the genetic variant of SPI1 in GWAS

study was a loss of function or gain of function and relate it to the results of the in vivo knockdown and transgenic model experiments in the paper.

Response: To address this question, we now discussed the implication of our finding in the context of the SPI1 genome-wide association study (GWAS) in the discussion section.

- Page 20, line 427-443: In the discussion section, we added the following sentences “As we mentioned earlier, recent GWAS studies have identified *SPI1* as a genetic risk factor for AD^{6,7,8}. These strong genetic associations further increase the potential of *SPI1* as a therapeutic target in AD. Among the genetic variants linked to AD at the *SPI1* locus, the SNP rs1057233 is known as a protective allele and was associated with lower *SPI1* expression in myeloid cells⁷. This intriguing inverse correlation between the protective allele and *SPI1* mRNA level suggests that reducing SPI1 expression might potentially offer protection against certain AD phenotypes. On the surface, this hypothesis appears to conflict with our functional data, which showed that amyloid pathology was exacerbated upon knockdown of *Spi1* (Figs. 1 and 2), whereas overexpressing *Spi1* significantly ameliorated these phenotypes (Figs. 3, 4, and 8). This apparent conflict might be explained by the difference in cell types. The rs1057233 SNP is associated with reduced *SPI1* mRNA levels in peripheral immune cells (monocytes and macrophages)⁷. Interestingly, a recent study demonstrated that *SPI1* has a low correlation in gene expression between peripheral monocytes and brain microglia⁹. Therefore, to directly address whether the *SPI1* variant contributes to the disease onset by a loss or gain of function mechanism, it would be necessary to use a mouse model harboring the *SPI1* variant in future studies.”

2. In Fig.4d-i, could authors describe any differences in microglia clusters or cell-cell communications between NTG and TG mice? The main theme of the study would be the effect of SPI1 on cellular response. The organization of the figure seems to be more focused on extracting microglia cluster (in here, M4, 8, and 10) that enriched in DAM like gene network or cell-cell communication between clusters.

Response: We identified DEGs between *Spi1*^{+/+};5XFAD (WT) and *Spi1*^{Tg/0};5XFAD (TG) mice for microglia clusters (Table S12 and Extended Data Fig. 8). In our new analysis, interestingly, when we investigated changes based on genotype, *Ccl3* signaling decreases in TG mice, and *Ccr5* signaling is significantly downregulated in clusters M4 and M5 in TG mice (Table S15). We now updated the text to reflect these additional findings in the manuscript. In addition, Table numbers have been revised to accommodate the inclusion of Table S15.

As the reviewer indicated, Figure 4h (currently updated it as Fig. 9e) indicates cell-cell communication pathways between Microglial Cluster 4 to all other microglial clusters in both genotypes. Then we focused on the CCL pathway across all cells as it showed the greatest differential inferred activity from microglia to other cells, and the expression of the ligands and receptors was sufficient between genotypes for us to be confident in the finding (Fig. 9f). We found that this pathway mostly signals from Microglial Cluster 4, Microglial Cluster 8 and Microglial Cluster 10, all of which exhibit DAM signature.

- Page 16, line 339-344: In the results section, we added “To further clarify the direction of communication between ligand and receptor, we focused on the expression of *Ccl3*, *Ccl4*, and *Ccr5*. As expected, *Ccl3* and *Ccl4* are expressed primarily by DAM clusters M4, M8, and M10, while *Ccr5* is expressed in nearly all microglia (Extended Data Fig. 10). Interestingly, *Ccl3* signalling decreases in *Spi1*^{Tg/0};5XFAD compared to *Spi1*^{+/+};5XFAD mice, and *Ccr5* is significantly downregulated in microglial clusters M4 and M5 in *Spi1*^{Tg/0};5XFAD mice (Table S15).”

3. In lane 129-133, what is an interpretation for upregulated royal-blue modules such as lipid metabolism, neuronal development and gamma-secretase proteolysis?

Transcription factor Spi1 would primarily affect microglia or other myeloid cells. Unlike the result from proteomics data, authors checked beta-secretase expression and end products in extended Fig.5. What is the

author's opinion on which cells' gammasecretase function in brain tissue would be regulated by changes in *Spi1* gene expression and how?

Response: We appreciate the reviewer's insightful question. The pathway enrichment analysis of royal-blue modules is based on WGCNA analysis. This analysis focuses on identifying comparisons between weighted and unweighted correlation networks rather than individual differentially expressed proteins.

The Gamma-secretase proteolytic targets comprise 76 components, out of which 7 were detected in our data. Four of these components (Amyloid beta 42, Amyloid beta 40, APP-C83 (CTF), APP) were evaluated using biochemical analyses. Two of them, Amyloid beta 42 and Amyloid beta 40, exhibited significant upregulation upon *Spi1* knockdown mice compared to WT mice. These significant changes in some of the components may have led to the identification of annotations such as gamma-secretase proteolytic targets.

However, as strongly requested by Reviewer 1 (Q8), we now removed the proteomics data from the manuscript. Hence, our response to this question is not included in our revised manuscript.

4 & 5. From integrative transcriptomics analysis, authors extracted biological pathways such as cytokine, complement, immune responses (Fig.2, Fig.5). *C1q*, *Trem2*, *Tyrobp*, *Syk* genes are closely related each other. Their protein products have an important role in phagocytosis and microglia clustering. Did proteomics data also support their expression data which regulated by PU.1.?

In line with previous questions, the study lacks a biochemical data about A β clearance. Authors showed that *Spi1*-overexpression decreases A β burden (Fig1, Fig.3, and extended Fig.5). As authors confirmed that *Spi1* gene does not affect APP processing (extended Fig.3), it would be worth to check how did *Spi1* overexpression reduced A β burden and whether glial A β clearance function have changed.

Response: Following this insightful suggestion, we now performed an A β uptake (clearance) assay upon the regulation of *Spi1* expression to address this mechanistic question. *Spi1* knockdown significantly decreased the clearance of fibrillar A β (fA β), whereas *Spi1* overexpression increased fA β clearance in microglia. These data were also provided in response to the 1st reviewer's question (Q12) in this file. For the reviewers' convenience, we added them below again. We believe these new data now provide a critical mechanistic insight and appreciate the reviewer's insightful suggestion.

- Page 12, line 255-262: In the result section, we have added "To more directly investigate the role of *Spi1* in microglial response to aggregated A β , we performed fibrillar A β (fA β) uptake assays with fA β labeled with a pHrodo Red probe. BV-2 microglial cells were transfected with a *Spi1* siRNA or *Spi1* plasmid to knockdown or overexpression of *Spi1*, respectively. Twenty-four hours post-transfection, we evaluated a time-dependent A β uptake level after treatment with pHrodo-labeled fA β . *Spi1*-knockdown significantly decreased fA β uptake (Fig. 7e), while *Spi1*-overexpression increased (Fig. 7f). These findings suggest that *Spi1* levels modulate microglial A β clearance function, consequently altering microglial responses around amyloid plaques."

Based on the strong recommendation by reviewer #1, we now removed our proteomics data that was statistically underpowered.

6. In lanes 262-266, There are some doubts about the interpretation. "Additionally, we observed signaling from a small population of T cells to microglia through the same pathway", which suggests that T cells may also be involved in the amyloid pathology model. It is a surprise that small portions of T cells have observed in sequencing analysis (Fig. 4d). However, a recent article that referenced in this paper compares APOE4;APP/PS1 or APOE4;5xFAD and APOE4/TauP301S mouse models and provides data showing that amyloidosis model has very small numbers of T cells in parenchyma which comparable with APOE4;WT mice, but tauopathy model has significantly increased numbers of T cells. The literatures used to make curation in these types of analyses will also include studies performed in peripheral tissues.

Therefore, it can be misleading to make claims based on analytics data alone. Since this is far from the main issue of the paper, authors can simply tone down the discussion. If the authors want to make a point, they should perform some biological validation, such as staining, as shown in the references.

Response: We agree with the reviewer's thoughtful comment. The involvement of T-cells in 5XFAD pathology is beyond the scope of this paper, and the low number of cells in the data precludes any in-depth analysis. Therefore, we now removed the discussion of T-cells from the manuscript.

Minor comments

1. The group names in the figure and the manuscript don't match well. For example, abbreviation WT means *Spi1*^{+/+};APP/PS1 mice. I understand that it's a wild type for *Spi1*, but it can be confusing. Then NTG should be displayed as WT as well, or just describe them both as *Spi1*^{+/+};5XFAD (extended Fig.3). Authors should use official group names as they used in the manuscript or use more appropriate abbreviations.

Response: We agree with the reviewer that accurately designating genotypes was important because we used both *Spi1*-knockdown and *Spi1*-overexpression approaches in this study. That is why we used the abbreviation of WT for *Spi1*^{+/+};APP/PS1 (wild type for *Spi1*) and NTG for *Spi1*^{0/0};5XFAD (non-transgenic for *Spi1*) to distinguish the wild type mice used in each approach.

However, as the reviewer pointed out, we realized that this could be confusing, so we changed it to *Spi1*^{+/+};5XFAD (WT) from *Spi1*^{0/0};5XFAD (NTG) as suggested by the reviewer. To ensure consistency and clarity, we used the following system to describe the groups: **1)** *Spi1* knockdown approach; *Spi1*^{+/+};APP/PS1 (WT) and *Spi1*^{+/-};APP/PS1 (KD), **2)** *Spi1* overexpression approach, *Spi1*^{+/+};5XFAD (WT) and *Spi1*^{Tg/0};5XFAD (TG). We maintained the same group names throughout the figure and text. The graphs are labeled as either WT and KD; WT and TG for each approach due to space limitations.

2. In Fig.4e and extended Fig.8, it is curious that microglia cluster M4,8 and 10 are enriched with DAM genes but Fig.4e lacks M10 cluster information. Some of the same genes repetitively observed in multiple modules in chord plot (Fig.4h). Are some clusters significantly different enough to be categorized separately?

Response: Figure 4e (currently updated it as Fig. 9b) is a heatmap that illustrates the differentially expressed gene (DEG) ontology for each microglia cluster. As the reviewer pointed out, cluster M10 is not included in Fig. 9b. The reason for this exclusion is the small number of cells in this cluster. After correcting for multiple comparisons, only three genes were found to be differentially expressed in this cluster, and these genes do not exhibit significant enrichment in any KEGG ontologies. For this reason, we have not shown M10 in Fig. 9b.

The reviewer mentioned some genes that are "repeatedly observed in multiple modules of the chord plots" in Fig. 9e. Our labelling M1 is being confused for Module 1 rather than Microglial Cluster 1. Therefore, we now revised the figure to clarify this. For further clarification, *Ccl3* and *Ccl4* are both expressed in microglia cluster 4 (purple) and signal to *Ccr5* in multiple other microglial clusters, hence *Ccr5* being shown several times. Because these genes are not selective markers for a particular microglia cluster, they can show up in different clusters. This data is now shown in the updated manuscript Fig. 9, which needed to be rearranged because we addressed the reviewer's response.

Updated Fig 9e

3. “Effects of~” term is somewhat neutral to be used in the title. Authors should better select other words to write a title that captures the conclusion of the study.

Response: We completely agree with the reviewer’s critique. Because we utilized both knockdown and overexpression mouse models, we also wanted to use a title that explicitly indicates the use of two models and the direction of the effects on AD phenotypes, with knockdown of SPI1 being detrimental whereas overexpression of it being beneficial. However, we could not do so due to a restrictive requirement by the journal. Up to 15 words were allowed and the title “should not contain active verbs” per the journal’s guideline.

Reviewer #3 (Remarks to the Author):

Recent work has implicated Spi1, which encodes PU.1, as a genetic risk factor for Alzheimer’s disease (AD). Spi1 is primarily expressed by microglia in the brain and there has been a few studies which have elucidated how it might play a role in AD. However, these studies have been in vitro and to my knowledge there has not been an in vivo study modulating the levels of Spi1 in an amyloid model. In this manuscript, Kim et al attempts to address this gap in the literature. First, they crossed APP/PS1-21 mice to *Spi1*^{-/-} to obtain *Spi1*^{-/+}; APP/PS1-21 (KD) or *Spi1*^{+/+};APP/PS1-21 (WT) mice in order to understand the effect of knocking down Spi1 on amyloidosis. They found that KD mice had significantly more amyloid than WT mice by several methodologies. They then performed Nanostring gene analysis, proteomics, and subsequent pathway analyses to determine by what mechanism Spi1 modulation impacts amyloidosis. According to their analysis, they determined that Spi1 KD resulted in a reduction in microglial phagocytosis, and this was potentially responsible for the increase in amyloidosis. Next, they crossed 5XFAD mice to Spi1Tg/0 to generate either Spi1Tg/0; 5XFAD (TG) or 5XFAD with non-transgenic Spi1 (NTG) to investigate the effect of overexpression of Spi1 on amyloidosis. Consistent with their previous experiment, they found that overexpression of Spi1 lowered amyloidosis. They then performed both Nanostring gene analysis and single-cell RNA sequencing (scRNAseq) comparing NTG and TG mouse cortices. According to their Nanostring analysis, the microglial phagocytosis pathway was surprisingly decreased

in TG mice compared to NTG mice. In their scRNAseq experiment, they captured 16,456 cells that primarily contained microglia. They were able to identify 19 clusters, 11 of which were microglia. They found that cluster 4 was particularly enriched in DAM markers. Using the CellChat program, they found that microglial cluster 4 may signal to other microglial clusters through Ccl3 and Ccl4 by the Ccr5 receptor. In looking at the DEGs from the KD and overexpression experiments together, they found that 6 genes change transcriptionally in opposite directions. Lastly, they found that Spi1 overexpression reduces gliosis and decreases dystrophic neurites.

Although this study addresses an important question and the amyloid, gliosis, and dystrophic neurite quantifications are sound, all meeting the standards of Nat. Comm., there seems to be several methodological problems in the multi-omic analyses that limit the ability to draw conclusions into the mechanism whereby Spi1 modulation alters amyloidosis.

Major comments

1. For the KD vs WT Nanostring experiment in Figure 2, none of the “DEGs” p -adj are <0.05 (Supplementary table 2). All the genes called “DEGs” are based on the raw p -values, which are not corrected for multiple testing (FDR adjusted p -values). It is unclear whether all these DEGs might just be false positives. If the authors believe these changes aren't false positives, they should prove it using another method such as qPCR. Also, this caveat should be explicitly stated in the paper, so readers can make their own conclusions about the data without assuming that the DEGs are derived from FDR corrected p -values.

Response: We acknowledge the concern raised regarding the use of raw p -values and the potential for false positives in our Nanostring experiment. We also appreciate constructive suggestions to validate our candidate DEGs using an independent method, such as qRT-PCR.

In our submission, we did not apply the conservative Bonferroni multiple comparisons test because it has been common practice in the field not to use it for the Nanostring dataset (unlike RNA-seq data). The guideline provided by the Nanostring company also recommends **not** to use the conservative Bonferroni multiple comparisons test. Because the genes included in the targeted CodeSet panel are known or suspected to be affected by the biology under study, their expression tends to correlate with each other more often than random genes not in the related pathways. Therefore, the resulting expression correlations in gene expression levels actually **violate** the statistical assumption of the independence of significance levels between tests.

Most importantly, following the reviewer's suggestion, we now performed a qPCR analysis to validate the expression levels of several DEGs between the cortex of $Spi1^{+/-};APP/PS1$ and $Spi1^{+/+};APP/PS1$ mice. The mRNA levels of *Tyrobp*, *Tcirg1*, and *Cd74* were significantly upregulated in $Spi1^{+/-};APP/PS1$ mice compared to $Spi1^{+/+};APP/PS1$ mice, whereas the mRNA levels of *Hspa1b* and *Pard3* were significantly downregulated. We now added the new figure shown below to the Fig. **5d** and updated the result section with these additional new data, accordingly.

Updated Fig. 5a-d

- Page 9-10, line 193-199: In the results section, we now added “We validated those DEGs involved in the TYROBP Causal Network (*Tyrobp* and *Tcigr1*) and Immune response_Antigen presentation by MHC class I (*Hspa1a*, *Cd74*, and *Tyrobp*), and DEG such as *Pard3* in the cortex of *Spi1*^{+/-};APP/PS1 mice and *Spi1*^{+/+};APP/PS1 mice using qPCR analysis. Consistent with the NanoString data (Fig. 5a), *Tyrobp*, *Tcigr1*, and *Cd74* mRNA levels were significantly upregulated in *Spi1*^{+/-};APP/PS1 mice compared to *Spi1*^{+/+};APP/PS1 mice, whereas *Hspa1b* and *Pard3* were significantly downregulated (Fig. 5d).”

2. Most of the pathways that are “enriched” in Figure 2B-C have 1-2 genes out of the pathway (Table 3) and due to the issue addressed above it is arguable whether these genes have altered expression when *Spi1* is KD.

Response: To address the reviewer's concerns, we now performed qPCR analysis of DEGs involved in the TYROBP Causal Network (*Tyrobp* and *Tcigr1*) and Immune response_Antigen presentation by MHC class I (*Hspa1a*, *Cd74*, and *Tyrobp*) pathways. The mRNA level of those DEGs can be found in the response to the same reviewer's Q1.

3. The proteomics data in Figure 2 is hard to interpret. There is no volcano plot showing differentially expressed proteins. As far as I can tell, there are no stats comparing proteins found between conditions in the supplementary table 5-6. There is only the WGCNA analysis which is difficult to interpret without knowing which proteins were changed by *Spi1* KD. The authors do go on to identify proteins that are in the royal blue enriched module, but it is unclear whether these proteins were actually changed significantly by *Spi1* KD or they are just in the overall module which was changed. In other words, the authors focus on changes in WGCNA modules but do not disclose which proteins in these modules were changed by *Spi1* KD.

Response: As strongly requested by Reviewer 1 (Q8), we now removed the proteomics data from the manuscript. Hence, our response to this question is not included in our revised manuscript.

4. For the TG vs NTG Nanostring experiment in Figure 4, only 2 of the “DEGs” *p*-adj are <0.05 (Supplementary table 12). As mentioned above, the genes with a raw *p*-value <0.05 but not with a FDR corrected *p*-value <0.05 might just be false positives. If the authors believe these changes aren't false positives, they should prove it using

another method such as qPCR. Also, this caveat should be explicitly stated in the paper, so readers can make their own conclusions about the data without assuming that the DEGs are derived from FDR corrected p-values.

Response: This question was addressed in the response to the same reviewer's Q1. To avoid duplication, please refer to the response in Q1.

In addition, responding to the reviewer's suggestion, we performed qPCR analysis on several DEGs between the cortex of *Spi1^{Tg/0}*;5XFAD mice and *Spi1^{+/+}*;5XFAD mice to validate their mRNA levels. The mRNA levels of *C1qa*, *Fcer1g*, *Tyrobp*, *Trem2*, *Cyba*, *Ctss*, and *Laptm5* were significantly downregulated in *Spi1^{Tg/0}*;5XFAD mice compared to *Spi1^{+/+}*;5XFAD mice, whereas the mRNA levels of *Hspa1b* was significantly upregulated. We now added the figure below in the Fig. 5h and updated the result section with these additional new data, accordingly.

Updated Fig. 5e-h

- Page 10, line 211-216: In the results section, we now added “Next, we also validated several DEGs involved in the Microglia pathogen phagocytosis pathway (*C1qa*, *Fcer1g*, *Tyrobp*, *Trem2*, and *Cyba*), and DEGs such as *Ctss*, *Laptm5*, and *Hspa1a* in the cortex of *Spi1^{Tg/0}*;5XFAD mice and *Spi1^{+/+}*;5XFAD mice using qPCR analysis. Consistent with the NanoString data (Fig. 5e), *C1qa*, *Fcer1g*, *Tyrobp*, *Trem2*, *Cyba*, *Ctss*, and *Laptm5* mRNA levels were significantly downregulated in *Spi1^{Tg/0}*;5XFAD mice compared to *Spi1^{+/+}*;5XFAD mice, whereas *Hspa1b* was significantly upregulated (Fig. 5h).”

5. For the single cell experiment, it should be explicitly stated how many biological replicates the data is derived from and how many cells came from which animal. I could not find that information and without it the data from the experiment is hard to interpret.

Response: We apologize for missing this information. This question was addressed in the 1st reviewers’ response Q10 in this file. For the reviewers’ convenience, we have addressed this one more time.

Page 37, line 889-892: In the methods section, we now added “Single-cell suspensions were prepared from the brains of two *Spi1^{+/+}*;5XFAD and two *Spi1^{Tg/0}*;5XFAD mice, yielding 16,532 cells from *Spi1^{+/+}*;5XFAD mice and 18,247 cells from *Spi1^{Tg/0}*;5XFAD mice. These suspensions were processed in a single batch by the 10X Chromium.”

6. The analysis of the single cell experiment seems to be mostly limited to examining microglial markers in the integrated dataset (containing TG and NTG cells) rather than making any comparisons between the two. There are no proportion differences in clusters shown between the TG and NTG groups. Supplementary table S15 is labeled “Cell-type clusters DEGs between *Spi1^{Tg/0}*;5XFAD and *Spi1^{+/+}*;5XFAD mice.” I believe that this is actually the DEGs (which are FDR corrected) between all the clusters for the integrated dataset rather than a comparison of DEGs between TG and NTG for each cluster. This seems to correspond with extended data figure 6. Please make this clearer if this interpretation is wrong and these are actually differentially expressed genes between the TG and NTG conditions. Without direct comparisons between the TG and NTG groups, it is unclear what the reader should derive from this information that addresses the scientific question posed about the relationship between *Spi1* overexpression and amyloidosis.

Response: This question was addressed in the 1st reviewers’ response Q10 in this file. For the reviewers’ convenience, we have addressed this one more time below.

To address the reviewer’s question about relative cell proportions from each genotype within each cluster, we employed Propellor, a Bayesian tool for analyzing changes in cell proportions per cluster per condition. While we observed variations in proportions between cells per cluster per genotype, none of these differences reached statistical significance (*p*-values ranging from 0.973 to 0.928). We now added the figure below in the Expanded Data Fig. 7 and stated that “There were no significant differences in cell populations within each cluster between the two genotypes (Extended Data Fig. 7 and lines 291-292)”.

Ratio of average cell proportions per cluster per genotype. Shown are bar plots per each cell cluster (M1-11=microglia; A1-2=astrocytes; En=endothelial cells; Er=erythrocytes; Ma=macrophages; N1=neurons; O=oligodendrocytes; and T=T-cells) representing the ratio of average cell proportions between the genotypes (Cells in *Spi1^{Tg/0}*;5XFAD mice/Cells in *Spi1^{+/+}*;5XFAD mice). *Benjamini-Hochberg*-adjusted *P*-values are shown at the end of each bar. *P*-values are shown at the end of each bar.

Supplemental Table S15 (now updated to Table **S12**) is indeed DEGs by cluster between *Spi1*^{+/+};5XFAD and *Spi1*^{Tg/0};5XFAD mice. To clarify, we wrote as shown below.

- Page 14, line 293-301: In the results section, we stated “To gain more insight into the potential pathways regulated by *Spi1*-overexpression, we identified DEGs between *Spi1*^{Tg/0};5XFAD and *Spi1*^{+/+};5XFAD mice for each cell-type cluster (Table **S12** and Extended Data Fig. **8**). Astrocyte clusters 1-2, Neurons, and Microglia clusters 1-10 exhibited significant changes in gene expression caused by *Spi1*-overexpression. After applying a 1.5-fold-change threshold, we identified 16 DEGs in astrocytes, 4 DEGs in neurons, and 27 DEGs in microglia (Table **S12**). In microglia, several genes were differentially expressed in the same direction across multiple microglia clusters. For example, *Camk2a*, *Camk2n1*, *H2-D1*, *Olfml3*, and *Qpct* were upregulated in microglial clusters 4 and 10, while *H2-DMa* was downregulated in microglial clusters 1,2,3,4,5,6, and 9 (Table **S12**).”

7. Without knowing the expression levels of *Ccl3/4* and *Ccr5* in the NTG and TG conditions, it is unclear how the CellChat findings in Figure 4 are relevant to the scientific question at hand.

Response: We identified DEGs between *Spi1*^{+/+};5XFAD (WT) and *Spi1*^{Tg/0};5XFAD (TG) mice for microglia clusters (Table **S12** and Extended Data Fig. **8**). In our new analysis, interestingly, when we investigated changes based on genotype, *Ccl3* signaling decreases in TG mice, and *Ccr5* signaling is significantly downregulated in clusters M4 and M5 in TG mice (Table **S15**). We now updated the text to reflect these additional findings in the manuscript. In addition, Table numbers have been revised to accommodate the inclusion of Table **S15**.

In addition, we demonstrated that *Ccl3/4* are expressed primarily by DAM clusters M4, M8, and M10, while *Ccr5* is expressed in nearly all microglia (Extended Data Figure **10b**).

- Page 16, line 339-345: In the results section, we added “To further clarify the direction of communication between ligand and receptor, we focused on the expression of *Ccl3*, *Ccl4*, and *Ccr5*. As expected, *Ccl3* and *Ccl4* are expressed primarily by DAM clusters M4, M8, and M10, while *Ccr5* is expressed in nearly all microglia (Extended Data Fig. **10**). Interestingly, *Ccl3* signalling decreases in *Spi1*^{Tg/0};5XFAD compared to *Spi1*^{+/+};5XFAD mice, and *Ccr5* is significantly downregulated in microglial clusters M4 and M5 in *Spi1*^{Tg/0};5XFAD mice (Table **S15**). Therefore, our data suggest that DAM-like microglia use the CCL pathway to communicate with non-DAM microglia in our models.”

Minor comments

1. It would be interesting to provide data on plaque size for the KD and overexpression studies since that could be easily calculated. Do the changes to microglia affect only plaque number or also plaque size?

Response: We appreciate this insightful question. We now analyzed the plaque size for the *Spi1* KD and OE data sets. We observed that changes in *Spi1* level exhibited alterations in both the number and size of amyloid plaques. *Spi1* KD mice showed an increase in amyloid plaque load and number of plaques (Figure **1e-i**), as well as increased overall plaque size (Extended Data **Figs. 2g**) compared to control mice. Conversely, *Spi1* TG mice exhibited a reduction in amyloid plaque load and number of plaques (Figure **3e-i**), as well as amyloid plaque size (Extended Data **Figs. 4e**) compared to control mice. We now added the following data to Extended Data Figures **2g** or **4e** and updated the main manuscript.

Updated Extended Data Fig. **2g** (left panel) and **4e** (right panel)

- Page 5, line 93-96: In the result section, we added “Similarly, the number of Aβ plaques was significantly increased in the cortex and hippocampus of *Spi1*-knockdown mice (Fig. 1h,i). In addition, the overall size of Aβ plaques were enlarged (Extended Data Fig. 2g).”

- Page 8, line 153-156: In the result section, we added “Furthermore, the number of Aβ plaques was also significantly decreased in female *Spi1^{Tg/0}*;5XFAD mice compared to *Spi1^{+/+}*;5XFAD mice (Fig. 3h,i), accompanied by a further reduction in the overall size of Aβ plaques (Extended Data Fig. 4e).”

2. Based on the proteomic WGCNA analysis, there seems to be little overlap with the Nanostring analysis. Although this is not surprising given the accumulating studies that show poor correlation between transcriptomics and proteomics, this is not acknowledged in the paper and the authors should try to speculate as to the reason for this.

Response: WGCNA analysis in proteomics focuses on identifying comparisons between weighted and unweighted correlation networks, rather than pinpointing differentially expressed individual proteins between genotypes. Therefore, we did not expect an overlap between proteomic WGCNA and Nanostring analyses. However, as strongly requested by Reviewer 1 (Q8), we now removed the proteomics data from the manuscript. Hence, our response to this question is not included in our revised manuscript.

3. Different amyloid mouse models used for KD (APP/PS1-21) and overexpression (5XFAD) experiments may limit the comparison of the experiments and could be the reason only 6 genes seemed to go in opposite directions from the KD and overexpression datasets.

Response: We agree with the reviewer that using different APP transgenic mouse models might have contributed to the limited overlap. Another point to consider is that the NanoString experiment is a “targeted” transcriptome analysis, not a whole transcriptome analysis. We identified 26 DEGs in the *Spi1* KD mice and 43 DEGs in the *Spi1* TG mice, of which 6 DEGs moved to opposite directions (23% and 14% respectively), which is not a trivial portion.

Background info: Our transition from APP/PS1-21 amyloid mouse model to the 5xFAD amyloid mouse model was mainly due to an administrative/logistical issue, not a scientific concern. Specifically, we had previously secured an MTA agreement for APP/PS1-21 mouse model when our lab was located at the Mayo Clinic, Jacksonville, FL for the SPI1 knockdown mouse project. However, due to the absence of an MTA agreement at Indiana University where our lab was later relocated, we opted to generate PU.1 OE mice on the 5xFAD background.

Most importantly, we thought using two different APP mouse models would even increase the rigor of our study. In fact, our findings demonstrate that *Spi1*-mediated effects are not limited to only one particular APP mouse model. We believe obtaining consistent data across two different APP mouse models is one of the main strengths of our manuscript.

- Page 11, line 221-223: In the methods section, we revised “We utilized a linear model¹⁰ of gene expression against genotype to identify DEGs and then performed analyses of Transcription factor enrichment, Biological Pathways, GO processes, and Process Networks using the combined DEG list (Fig. 6a).”

References

1. Rosenbauer F, *et al.* Lymphoid cell growth and transformation are suppressed by a key regulatory element of the gene encoding PU.1. *Nat Genet* **38**, 27-37 (2006).
2. McKercher SR, *et al.* Targeted disruption of the PU.1 gene results in multiple hematopoietic abnormalities. *Embo j* **15**, 5647-5658 (1996).
3. Sierksma A, *et al.* Novel Alzheimer risk genes determine the microglia response to amyloid-beta but not to TAU pathology. *EMBO Mol Med*, e10606 (2020).
4. Rustenhoven J, *et al.* PU.1 regulates Alzheimer's disease-associated genes in primary human microglia. *Mol Neurodegener* **13**, 44 (2018).
5. Gupta N, Jadhav S, Tan K-L, Saw G, Mallilankaraman KB, Dheen ST. miR-142-3p Regulates BDNF Expression in Activated Rodent Microglia Through Its Target CAMK2A. *Frontiers in cellular neuroscience* **14**, (2020).
6. Gjonjeska E, *et al.* Conserved epigenomic signals in mice and humans reveal immune basis of Alzheimer's disease. *Nature* **518**, 365-369 (2015).
7. Huang KL, *et al.* A common haplotype lowers PU.1 expression in myeloid cells and delays onset of Alzheimer's disease. *Nature neuroscience*, (2017).
8. Kunkle BW, *et al.* Genetic meta-analysis of diagnosed Alzheimer's disease identifies new risk loci and implicates Abeta, tau, immunity and lipid processing. *Nat Genet* **51**, 414-430 (2019).
9. Lopes KDP, *et al.* Genetic analysis of the human microglial transcriptome across brain regions, aging and disease pathologies. *Nature Genetics* **54**, 4-17 (2022).
10. Ritchie ME, *et al.* limma powers differential expression analyses for RNA-sequencing and microarray studies. *Nucleic Acids Research* **43**, e47-e47 (2015).

Reviewers' Comments:

Reviewer #1:

Remarks to the Author:

The revised manuscript is now far more readable and interpretable. All of my previous concerns have been addressed. The authors should be commended for such thorough and meticulous work (I only wish all manuscripts in the AD field met this level of rigor!).

Reviewer #2:

Remarks to the Author:

The part that the reviewer had questioned has been corrected well. However, the regrettable part is that all the proteomics data pointed out by the reviewer has been removed. If that part had been revised and improved a little more, the paper would have been more meaningful. Now that the controversial part has been removed, there are no further comments on the current manuscript status.

Reviewer #3:

Remarks to the Author:

The authors have adequately address the comments of the previous review.